# REDEFINING THE SELF-NORMALIZATION PROPERTY

## ABSTRACT

The approaches that prevent gradient explosion and vanishing have boosted the performance of deep neural networks in recent years. A unique one among them is the self-normalizing neural network (SNN), which is generally more stable than initialization techniques without explicit normalization. The self-normalization property of SNN in previous studies comes from the Scaled Exponential Linear Unit (SELU) activation function. However, it has been shown that in deeper neural networks, SELU either leads to gradient explosion or loses its self-normalization property. Besides, its accuracy on large-scale benchmarks like ImageNet is less satisfying. In this paper, we analyze the forward and backward passes of SNN with mean-field theory and block dynamical isometry. A new definition for self-normalization property is proposed that is easier to use both analytically and numerically. A proposition is also proposed which enables us to compare the strength of the self-normalization property between different activation functions. We further develop two new activation functions, leaky SELU (lSELU) and scaled SELU (sSELU), that have stronger self-normalization property. The optimal parameters in them can be easily solved with a constrained optimization program. Besides, analysis on the activation's mean in the forward pass reveals that the self-normalization property on mean gets weaker with larger fan-in, which explains the performance degradation on ImageNet. This can be solved with weight centralization, mixup data augmentation, and centralized activation function. On moderate-scale datasets CIFAR-10, CIFAR-100, and Tiny ImageNet, the direct application of lSELU and sSELU achieve up to 2.13% higher accuracy. On Conv MobileNet V1 - ImageNet, sSELU with Mixup, trainable $\lambda$, and centralized activation function reaches 71.95% accuracy that is even better than Batch Normalization.

## 1 INTRODUCTION

In recent years, deep neural networks (DNNs) have achieved state-of-the-art performance on different tasks like image classification (He et al., 2015; Zheng et al., 2019). This rapid development can be largely attributed to the initialization and normalization techniques that prevent the gradient explosion and vanishing. The initialization techniques (He et al., 2015; Xiao et al., 2018) initialize the parameters in networks to have good statistical property at beginning, and assume that this property can be more or less maintained throughout the training process. However, this assumption is likely to be violated when the network gets deeper or is trained with higher learning rate. Hence, normalization techniques are proposed to explicitly normalize the network parameters (Salimans & Kingma, 2016; Arpit et al., 2016) or the activations (Ioffe & Szegedy, 2015b; Ulyanov et al., 2016) during training. In particular, Batch Normalization (BN) (Ioffe & Szegedy, 2015a) has become a standard component in DNNs, as it not only effectively improves convergence rate and training stability, but also regularizes the model to improve generalization ability.

However, BN still has several drawbacks. First, when calculating the mean and variance, the accumulation must be done under FP32 to avoid underflow (Micikevicius et al., 2018). This brings challenges when training neural networks in low bit width. Second, the performance degradation under micro batch size also makes it more difficult to design training accelerators, as the large batch size increases the memory size to store the intermediate results for backward pass (Deng et al., 2020). Besides, Chen et al. (2020a); Wu et al. (2018) show that BN introduces considerable overhead.

The self-normalizing neural network (SNN) provides a promising way to address this challenge. SNN initializes the neural network to have a good statistical property at the beginning just like

initialization techniques. However, the statistics deviation in forward and backward passes can be gradually fixed during propagation, thus it is more robust to the deviation from initial properties (Chen et al., 2020b). For instance, the mean and variance of output activations with SELU in Klambauer et al. (2017) automatically converge the fixed point $(0, 1)$. Chen et al. (2020b) analyze the Frobenius norm of backward gradient in SNN activated with SELU. They reveal a trade-off between the self-normalization property and the speed of gradient explosion in the backward pass, and the hyper-parameters need to be configured according to the depth of the network. The resulting activation function, depth-aware SELU (dSELU), has achieved even higher accuracy than the original configuration on moderate-scale datasets like CIFAR-10, and makes the SNN trainable on ImageNet.

However, in deeper neural networks, the dSELU gradually degenerates to ReLU and loses its self-normalization property. Moreover, even with dSELU, the test accuracy on ImageNet with Conv MobileNet V1 (Howard et al., 2017) is still 1.79% lower than BN (Chen et al., 2020b). Therefore, we aim to answer the following three questions in this paper: 1). Is SELU the only activation function that has self-normalization property? 2). If it is not, is there a better choice? And how do we compare the strength of self-normalization property between different activation functions? 3). Why the performance of SNN on ImageNet is less satisfying? Is there any way to alleviate that?

In this paper, we analyze the signal propagation in both forward and backward passes in serial deep neural networks with mean-field theory (Poole et al., 2016) and block dynamical isometry (Chen et al., 2020b). Our main theoretical results are summarized as follows:

- We illustrate that an activation function would demonstrate self-normalization property if the second moment of its Jacobian matrix's singular values $\phi(q)$ is inversely proportional to the second moment of its input pre-activations $q$, and the property gets stronger when $\phi(q)$ gets closer to $1/q$. A new definition of the self-normalization property is proposed that can be easily used both analytically and numerically.

- We propose leaky SELU (lSELU) and scaled SELU (sSELU). Both of them have an additional parameter, $\beta$, that can be configured to achieve stronger self-normalization property. The hyper-parameters can be solved with a constrained optimization program, thus no additional hyper-parameter relative to dSELU is introduced.

- We reveal that models with larger fan-in have weaker normalization effectiveness on the mean of the forward pass signal. This can be solved with explicit weight centralization, mixup data augmentation (Zhang et al., 2018), and centralized activation function.

On CIFAR-10, CIFAR-100, and Tiny ImageNet, lSELU and sSELU achieves up to 2.13% higher test accuracy than previous studies. On ImageNet - Conv MobileNet V1, sSELU with Mixup, trainable $\lambda$, and centralized activation function achieves comparable test accuracy (71.95%) with BN. Besides, we provide a CUDA kernel design for lSELU and sSELU that has only 2% overhead than SELU.

## 2 RELATED WORK

In this section, we present an overview of existing studies on the self-normalizing neural networks (SNN) as well as statistical studies on forward and backward signals in deep neural networks.

**Self-normalizing Neural Network**. Scaled Exponential Linear Unit (SELU) (Klambauer et al., 2017) scales the Exponent Linear Unit (ELU) by a constant scalar $\lambda$. The $\lambda$ and original parameter $\alpha$ in ELU are configured such that the mean and variance of output activation have a fixed point (0, 1). The authors further prove that this fixed point is still stable and attractive even when the input activations and the weights are unnormalized. Chen et al. (2020b) investigate the fixed point in backward gradient. They reveal that the gradient of SNN is exploding with the rate $(1 + \epsilon)$ per layer, where $\epsilon$ is a small positive value. The self-normalizing property gets stronger when $\epsilon$ is larger, whereas the gradient will explode at a higher rate. Therefore, they propose the depth-aware SELU in which the $\epsilon \approx 1/L$ is used to derive the optimal $\alpha$ and $\lambda$ in SELU for a network with depth $L$.

**Statistical Analysis of Deep Neural Networks**. Schoenholz et al. (2016); Poole et al. (2016); Burkholz & Dubatovka (2018) investigate the forward activations under the limit of large layer width with mean-field theory. They have identified an Order-to-Chaos phase transition characterized by the second moment of singular values of the network's input-output Jacobian matrix. The neural network has good performance when it is on the border of the order and chaos phases. On the other

hand, Chen et al. (2020b) develop a very handy framework for analyzing the Frobenius norm of gradient. They illustrate that the gradient norm equality is a universal philosophy behind various different initialization, normalization techniques, and even some neural network structures. The gradient norm equality means the Frobenius Norm of the gradient is more or less equal in different layers so that the information flow in the backward pass can be well preserved. (Arpit & Bengio, 2020)

# 3 SELF-NORMALIZATION PROPERTY

In this section, we formally define the self-normalization property under the problem formulation, notations, and assumptions as follows.

**Problem Formulation**. Let's consider a DNN with $L$ layers. Each layer performs a linear transform followed by a non-linear element-wise activation function $f$, i.e.

$$\boldsymbol{x}_l = f(\boldsymbol{h}_l), \quad \boldsymbol{h}_l = \boldsymbol{W}_l \boldsymbol{x}_{l-1} + \boldsymbol{b}_l, \quad l = 1, ..., L, \tag{1}$$

where $\boldsymbol{x}_l \in \mathbb{R}^{N_l}$ is the output feature vector of layer $l$, $\boldsymbol{h}_l$ is the pre-activation vector, $\boldsymbol{W}_l$ is the weight of fully-connected layer or the expanded doubly block circulant matrix (Sedghi et al., 2019) of 2D convolution, $\boldsymbol{b}_l$ is the vector of biases, and we denote the loss as $\mathcal{L}$. Besides, without loss of generality, for $f$ and $x \sim N(0, q)$, we have

$$(1 + \delta_q) E\left[f^2(x)\right] = E[(df(x)/dx)^2] E[x^2], \tag{2}$$

where $\delta_q$ is a function of $q$. Following previous studies (Poole et al., 2016; Chen et al., 2020b), for $\forall\, l$, we make the assumptions as follows:

**Assumption 1** *The mean of entries in $\boldsymbol{W}_l$ and $\boldsymbol{b}_l$ are zero.*

**Assumption 2** *With central limit theory, the entries in $\boldsymbol{h}_l$ follow i.i.d. $N(0, q_l)$, $q_l = \frac{1}{N_l}\boldsymbol{h}_l^T\boldsymbol{h}_l$.*

**Assumption 3** *The eigenvalues of $\boldsymbol{W}_l^T\boldsymbol{W}_l$ are independent with entries in $\boldsymbol{h}_{l-1}$.*

Klambauer et al. (2017) first define the self-normalization property of a neural network as follows.

**Definition 1** *(Self-normalizing Neural Network) A neural network is self-normalizing if it possesses a mapping $g : \Omega \to \Omega$ for each activation $y$ that maps mean and variance from one layer to the next and has a stable and attracting fixed point depending on $(\omega, \tau)$ in $\Omega$. Furthermore, the mean and the variance remain in the domain $\Omega$, that is $g(\Omega) \subseteq \Omega$, where $\Omega = \{(\mu, \nu)|\mu \in [\mu_{min}, \mu_{max}], \nu \in [\nu_{min}, \nu_{max}]\}$. When iteratively applying the $g$, each point within $\Omega$ converges to this fixed point.*

This definition imitates the explicit normalization techniques like BN, which ensures that the feed-forward signal is normalized. Based on Definition 1, Klambauer et al. (2017) propose the SELU:

$$f(x) = \lambda \left\{ \begin{array}{ll} x & if\ x > 0 \\ \alpha e^x - \alpha & if\ x \leq 0 \end{array} \right. . \tag{3}$$

Besides, Klambauer et al. (2017) initialize the entries in $\boldsymbol{W}_l$ with $N(0, 1/N_{l-1})$, so that the output pre-activation will have the same second moment of input activation. With the stable fixed points of mean and variance around $(0, 1)$, the optimal choice for $\lambda$ and $\alpha$ can be derived from

$$\int_{-\infty}^{\infty} f(z) \frac{e^{-\frac{z^2}{2}}}{\sqrt{2\pi}} dz = 0, \quad \int_{-\infty}^{\infty} f^2(z) \frac{e^{-\frac{z^2}{2}}}{\sqrt{2\pi}} dz = 1. \tag{4}$$

Furthermore, the authors prove that the fixed points for mean and variance are still attractive even when the statistical properties of the parameters in the neural network deviate from the initial setup.

However, the statistical fixed point in the forward pass doesn't necessarily lead to good dynamics of gradient. Chen et al. (2020b) analyze the Frobenius norm of the gradient in neural networks activated by SELU. With the same activation function shown in equation 3, their analysis shows that the optimal $\lambda$ and $\alpha$ can be configured by preserving the Frobenius norm of backward gradient and second moment of forward activations with equations as follows:

$$\int_{-\infty}^{\infty} \left(\frac{df(z)}{dz}\right)^2 \frac{e^{-\frac{z^2}{2}}}{\sqrt{2\pi}} dz = 1 + \epsilon, \quad \int_{-\infty}^{\infty} f^2(z) \frac{e^{-\frac{z^2}{2}}}{\sqrt{2\pi}} dz = 1. \tag{5}$$

where $\epsilon$ is a small positive constant, without which the only solution for equation 5 would be $\lambda = \sqrt{2}$ and $\alpha = 0$, and the activation function degenerates back to ReLU with the initialization technique proposed in He et al. (2015). Thus it will lose the self-normalization property. Conversely, a relatively large $\epsilon$ will bring stronger self-normalization property, but meanwhile make the Frobenius norm of gradient explode with rate $(1+\epsilon)$ per layer. Notably, the original configuration of SELU can be obtained by setting $\epsilon = 0.0716$. Therefore, Chen et al. (2020b) assert that having $\epsilon \approx \frac{1}{L}$ could bring a good trade-off between gradient norm stability and self-normalization property. Experiments on CIFAR-10 and ImageNet show that the new configuration results in higher accuracy.

Inspired by Chen et al. (2020b), we formally redefine the self-normalization property as follows:

**Definition 2** *(Self-normalization Property) Given an activation function $f$, we define operand $\phi$ as*

$$\phi(q) = \int_{-\infty}^{\infty} \left( \frac{df(\sqrt{q}z)}{d\sqrt{q}z} \right)^2 \frac{e^{-\frac{z^2}{2}}}{\sqrt{2\pi}} dz. \tag{6}$$

*If $f$ satisfies:*

$$\phi(1) = 1 + \epsilon, \quad \int_{-\infty}^{\infty} f^2(z) \frac{e^{-\frac{z^2}{2}}}{\sqrt{2\pi}} dz = 1, \quad \min(1, \frac{1}{q}) < \phi(q) < \max(1, \frac{1}{q}), \tag{7}$$

*then we say $f$ has the self-normalization property.*

While the first two equations in equation 7 are identical to equation 5 that constructs fixed-points for both the second moment of activations and the Frobenius norm of the gradient, the third one makes these fixed points attractive, as we have the proposition as follows.

**Proposition 3.1** *(Strength of Self-normalization Property) Under all the three Assumptions and Definition 2, we represent $\phi(q)$ as a linear interpolation between 1 and $1/q$ as follows.*

$$\phi(q) = \begin{cases} 1 + (1 - \gamma_{q<1})(1/q - 1) & q < 1 \\ 1/q + \gamma_{q>1}(1 - 1/q) & q > 1 \end{cases}. \tag{8}$$

*where $\gamma_q \in (0, 1)$ is a function of $q$. Then the following conclusions hold (Proof: Appendix A.2):*

- *The self-normalization property gets stronger when $\gamma_{q<1}$ and $\gamma_{q>1}$ get closer to $0$. In particular, we have $|\gamma_{q<1}| \approx |\gamma_{q>1}| \approx |\frac{d\phi(q)}{dq}|_{q=1} + 1|$ when $q$ is around $1$.*

- *For layer $l$, the gradient explodes under the rate $(1 + \delta_{q_l})$, i.e. $\Pi_{i=1}^{l}(1 + \delta_{q_{i-1}})E\left[||\frac{\partial \mathcal{L}}{\partial \boldsymbol{h}_l}||_2^2\right] = q_0 E\left[||\frac{\partial \mathcal{L}}{\partial \boldsymbol{h}_0}||_2^2\right]$.*

Proposition 3.1 is derived based on Assumption 1, whereas the mean of weight matrices may shift during training. Fortunately, Proposition 3.2 shows that the deviation of the mean of forward activations can also be normalized by simply multiplying with the weight matrix.

**Proposition 3.2** *(Normalization of Mean) Under the assumption that the entries in the weight matrix $w_{ij}$ are independent with the input activations, and their expectation has an upper bound $\mu$, i.e. $\forall i, j, \ E[w_{ij}] \leq \mu$. Then we say multiplication with the weight matrix normalizes the mean if $\mu < \frac{1}{N_{l-1}}$ holds, where $N_{l-1}$ is the fan-in of the current layer $l$. Moreover, the mean is scaled down by ratio smaller than $\mu N_{l-1}$. (Proof: Appendix A.3)*

## 4 NOVEL SELF-NORMALIZING ACTIVATION FUNCTIONS

Proposition 3.1 reveals that $f$ with $\phi(q)$ closer to $1/q$ may have stronger self-normalization property. Therefore, on the basis of SELU, we propose to add an additional hyper-parameter $\beta$ that can be configured to bring $\phi(q)$ closer to $1/q$ and encode other interesting properties. As demos, we find the following two activation functions are quite promising.

**Scaled Scaled Exponential Linear Unit (sSELU)**. The sSELU is defined as follows

$$f(x) = \lambda \begin{cases} x & if \ x > 0 \\ \alpha e^{\beta x} - \alpha & if \ x \leq 0 \end{cases}. \tag{9}$$

The negative pre-activations are scaled by $\beta$ before fed into the activation function. This design is motivated by the observation that without the curvature provided by the exponential term $\alpha e^x$, $\phi(q)$ of SELU will be a constant value without self-normalization property.

**Leaky Scaled Exponential Linear Unit (lSELU)**. The lSELU is defined as follows

$$f(x) = \lambda \begin{cases} x & if \ x > 0 \\ \alpha e^x + \beta x - \alpha & if \ x \leq 0 \end{cases}, \tag{10}$$

which has an additional negative slope $\beta x$. This is inspired by the observation that leaky ReLU helps to avoid the saturation of negative activations. Besides, Chen et al. (2020b) show that leaky ReLU alone also improves the stability of the training process.

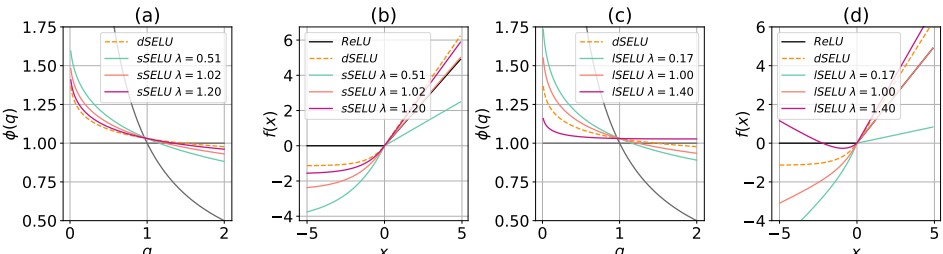

Figure 1: The $\phi(q) \sim q$ and $f(x) \sim x$ of sSELU (a) & (b) and lSELU (c) & (d) under different $\lambda$.

**Determine the optimal $\lambda$, $\alpha$, and $\beta$**. Figure 1 shows that given proper parameters $\lambda$, $\alpha$, and $\beta$, our sSELU and dSELU can be configured to get closer to $1/q$, which indicates stronger self-normalization property. With the first conclusion in Proposition 3.1 and equation 7, the $\lambda$, $\alpha$, and $\beta$ can be obtained by solving the optimization problem below when $\epsilon$ is provided:

$$\min_{\lambda,\alpha,\beta} \left| \left. \frac{d\phi(q)}{dq} \right|_{q=1} + 1 \right|, \quad s.t. \ \phi(1) = 1 + \epsilon, \quad \int_{-\infty}^{\infty} f^2(z) \frac{e^{-\frac{z^2}{2}}}{\sqrt{2\pi}} dz = 1, \ \lambda \geq 1. \tag{11}$$

In particular, the constraint $\lambda \geq 1$ is inspired by the argument in Klambauer et al. (2017) that "a slope larger than one can increase the variance if it is too small in the lower layer". In this paper, we find that constraining $\lambda \geq 1$ provides two other benefits. First, having $\lambda \approx 1$ helps to maintain the mean of the output activations around 0. Second, having larger $\lambda$ slows down the gradient explosion in the backward pass. The detailed discussion can be found in Appendix A.4.

**Determine the $\epsilon$**. While Chen et al. (2020b) propose to have $\epsilon < 1/L$ to avoid gradient explosion, where $L$ is the depth of the network, their derivation is based on the assumption that $\delta_q \approx 0$ in equation 2. However, after taking the nonzero $\delta_q$ into consideration, our Proposition 3.1 shows that the rate is actually $(1 + \delta_q)$ rather than $(1 + \epsilon)$. We plot the relationship between $(1 + \delta_q)$ and $q$ under different $\epsilon$ in Figure 2.

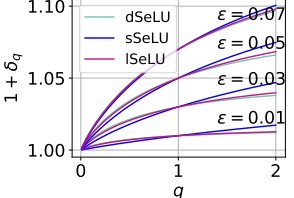

Figure 2: $(1 + \delta_q) \sim q$ of sSELU & lSELU under different $\epsilon$.

First of all, because of the first term in equation 7, we have $\delta_q = \epsilon$ when $q = 1$, this illustrates the intuition behind using $(1 + \epsilon)$ to characterize the rate of gradient explosion. Therefore, $\epsilon \approx 1/L$ is still a good choice to arbitrarily determine $\epsilon$, especially for relatively shallow networks. As lSELU and sSELU has relatively higher $\delta_{q>1}$ in Figure 2, a $\epsilon$ relatively smaller than that of dSELU may yield better performance. Last but not least, in deeper neural networks, $q$ has more chance to deviate from the fixed point $q = 1$, and $\delta_q$ gets larger when $q$ gets larger. Therefore, the trade-off between the strength of self-normalization property and the speed of gradient explosion may become too complex to be captured by $\epsilon \approx 1/L$, and it might be more promising to determine the proper $\epsilon$ on the validation set.

# 5 LARGE-SCALE SELF-NORMALIZING NEURAL NETWORK

Chen et al. (2020b) evaluate SELU and dSELU on Conv MobileNet V1 (Howard et al., 2017) on ImageNet. While SELU suffers from gradient explosion, the accuracy of dSELU is 1.79% lower than the BN baseline. We observe that there are two major reasons behind this performance degradation and propose several tricks that improve the performance on large-scale SNNs.

**Nonzero Mean in the Forward Pass**. Proposition 3.2 reveals that the nonzero mean can be diminished by multiplying with the weight matrices when $\mu < \frac{1}{N_{l-1}}$. On small-scale SNNs, as $N_{l-1}$ is relatively small, this condition is easy to satisfy, and we don't have to worry about the deviation of the mean from $0$. However, in large-scale SNNs for large datasets like ImageNet, larger fan-in is required to ensure the network has enough parameters to model the more complex problem. In Appendix A.5, we empirically show that models with larger fan-in tend to have larger $\mu N_{l-1}$, which implies weaker self-normalization property on the mean. As our Proposition 3.1 is based on Assumption 1, a greatly biased mean may violate the assumption. As a result, for large-scale SNNs, we have to consider the influence of nonzero-mean.

While the influence of the weight matrix on the mean is well captured by Proposition 3.2, the influence of the activation function is more complex. In particular, for layer $l$, we assume the pre-activations follow i.i.d. $N(E[h_l], \sigma^2)$, and the output mean can be computed with

$$E[x_l] = \int_{-\infty}^{\infty} f(x) \frac{1}{\sqrt{2\pi\sigma^2}} e^{-\frac{(x-E[h_l])^2}{2\sigma^2}} dx \tag{12}$$

We plot the relationship in Figure 3, in which the solid line represents the theoretical value and the dash line is the value measured via numerical experiments. When the variance $\sigma^2$ is large, there will be a positive bias on the mean of output. The explanation is quite intuitive: the saturated region in the negative axis has an asymmetric growth rate compared with the positive axis. Hence, when the variance is large, the positive part contributes more than the negative part, which increases the mean.

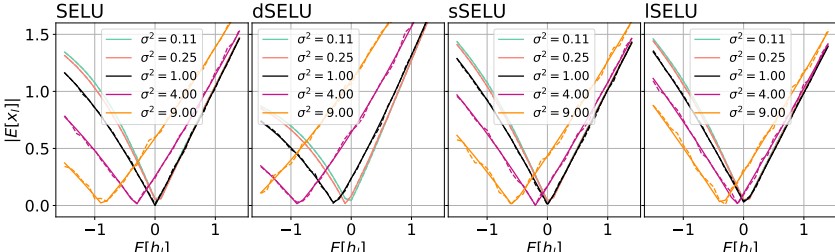

Figure 3: The absolute value of the output mean under different input mean and variance.

**Lack of Regularization during Training**. Luo et al. (2018) show that batch normalization also regularizes the training process. In particular, using the statistics of minibatch $\mu_B$ and $\sigma_B$ for explicit normalization introduces additional Gaussian noise that regularizes the training process, and it also discourages the reliance on a single neuron and penalizes correlations among neurons. However, activation functions with the self-normalization property don't have these features as they do not rely on the statistics from minibatch.

Based on the analysis above, we find that three techniques can be used to improve the performance of large-scale SNNs: mixup data augmentation (Zhang et al., 2018), weight centralization, and centralized activation functions.

**Mixup Data Augmentation**. The mixup is a simple data augmentation routine that constructs virtual training examples via linear interpolation: $\widetilde{x} = \gamma x_i + (1-\gamma)x_j, \widetilde{y} = \gamma y_i + (1-\gamma)y_j$, where $(x_i, y_i)$ and $(x_j, y_j)$ are two training samples randomly drawn from the training set, $\gamma \in (0, 1)$. In particular, we find that using Mixup with SNN brings two benefits.

First, Mixup reduces the variance / second moment of the inputs. Under the assumption that the corresponding entries $x_i$ and $x_j$ in the two samples are independent and $E[x_i^2] = E[x_j^2] := E[x^2]$, $E[x_i] = E[x_j] = 0$, we have $E\left[(\gamma x_i + (1-\gamma)x_j)^2\right] = (\gamma^2 + (1-\gamma^2))^2 E[x^2]$. For instance, when $\gamma = 0.7$, the second moment of the sample entries is 0.58 of the original training samples, hence the variance of the input samples is implicitly decreased. With a smaller $q$ in the first few layers, on

one hand, as shown in Figure 2, a smaller second moment $q$ leads to smaller $\delta_q$, which reduces the gradient explosion rate in the backward pass. On the other hand, as shown in Figure 3, a smaller variance also reduces the shift of output mean caused by the activation function.

Second, Mixup creates additional training samples from the dataset, which provides additional regularization that could further boost the accuracy. The same property is also used in Zhang et al. (2019). Besides, we empirically find that making $\lambda$ trainable is also helpful when applying lSELU and sSELU to large datasets like ImageNet. The trainable $\lambda$ can be viewed as the scalar multiplier initialized at 1 used in Zhang et al. (2019). Together with the bias of each layer, they serve as the affine transform applied in batch normalization (Ioffe & Szegedy, 2015a), which increases the representational power of the network (Michalski et al., 2019).

**Weight Centralization**. When $\mu < 1/N_{l-1}$, multiplication with the weight can effectively normalize the mean of activations. Therefore, we can explicitly centralize the weights, i.e $\hat{W} = W - mean(W)$. As the weights are usually much smaller than the feature maps, the overhead of Weight Centralization is usually quite small. Moreover, as it doesn't rely on the batch, Weight Centralization can still be utilized under micro-batch scenarios.

**Centralized Activation Function**. When the network with a large fan-in is relatively shallow, we can trade the strength of self-normalization property with the deviation of the mean caused by the activation function. While $\phi(1) = 1 + \epsilon$, $E[f(x)] = 0$, and $E[f^2(x)] = 1$ can not simultaneously hold in SELU and dSELU as they only have two parameters, the $\lambda$, $\alpha$, and $\beta$ in our sSELU and lSELU can be solved with

$$\phi(1) = 1 + \epsilon, \;\; E[f(x)] = \int_{-\infty}^{\infty} f(z) \frac{e^{-\frac{z^2}{2}}}{\sqrt{2\pi}} dz = 0, \;\; E[f^2(x)] = \int_{-\infty}^{\infty} f^2(z) \frac{e^{-\frac{z^2}{2}}}{\sqrt{2\pi}} dz = 1, \quad (13)$$

which ensures that the output activations still have zero-mean when the input is at the fixed point.

## 6 EXPERIMENTS

In this section, we validate our activation functions on multiple image classification benchmarks. In Appendix B, we present an efficient CUDA kernel design, under which the overhead of lSELU and sSELU are only 2% higher than SELU. The experiment setup is in Appendix C and the value of the parameters $\lambda$, $\alpha$, $\beta$, and the resulting $\gamma_{q=1}$ are summarized in Appendix D.

### 6.1 NORMALIZATION EFFECTIVENESS

We empirically show that our new activation functions have better normalization effectiveness than existing studies, which is demonstrated by the second moment of the output pre-activation of each convolutional layer ($E[h^2]$) and the Frobenius norm of the gradient of the weight ($||\frac{\partial \mathcal{L}}{\partial W}||_F$).

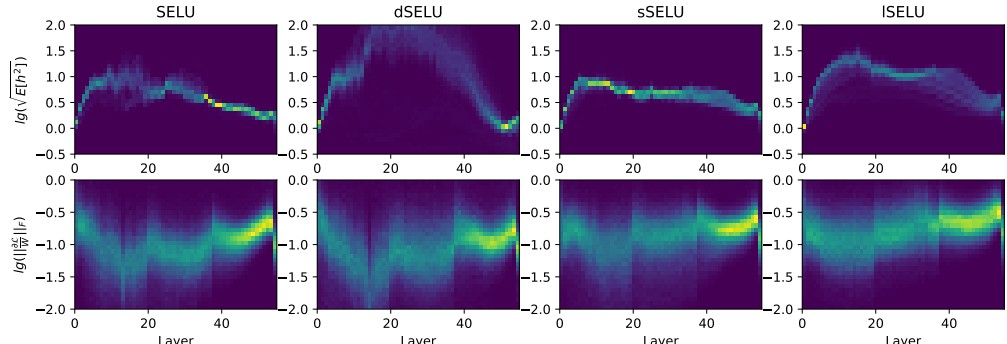

Figure 4: The distribution of the second-moment of forward pre-activation and the Frobenius norm of backward gradient on weights.

As shown in Figure 4, in the forward pass, sSELU and lSELU normalize the second moment in the forward pass better than dSELU. In the backward pass, compared with SELU and dSELU, both sSELU and lSELU have much flatter and more concentrated distribution of the Frobenius norm in the backward pass. Notably, SELU has $\epsilon \approx 0.0716$, and higher $\epsilon$ lead to stronger self-normalization property. This explains why it also has good dynamics in the forward pass. However, $\epsilon = 0.0716$

also increases the speed of gradient explosion, which explains why SELU has worse backward dynamics. Last but not least, the further the $E[h^2]$ deviates from 0 in the forward pass, the faster the $||\frac{\partial \mathcal{L}}{\partial W}||_F$ increases in the backward pass. As larger $q = E[h^2]$ will lead to larger $\delta_q$, this observation justifies the second conclusion in Proposition 3.1.

## 6.2 MODERATE-SCALE DATASETS

We summarize the results on CIFAR-10, CIFAR-100, and Tiny ImageNet in Table 1.

Table 1: Test accuracy on CIFAR-10, CIFAR-100, and Tiny ImageNet (Cl=95%).

| Method | $\epsilon$ | CIFAR-10 | CIFAR-100 | Tiny ImageNet |
|---|---|---|---|---|
| BN+ReLU | | $85.64\% \pm 1.15\%$ | $52.60\% \pm 1.98\%$ | $43.45\% \pm 0.58\%$ |
| SELU(Klambauer et al., 2017) | | $85.69\% \pm 1.10\%$ | $55.08\% \pm 0.37\%$ | $45.50\% \pm 0.47\%$ |
| dSELU(Chen et al., 2020b) | 0.01 | $86.07\% \pm 0.40\%$ | $54.09\% \pm 0.79\%$ | $45.63\% \pm 0.49\%$ |
| | 0.017 | $86.39\% \pm 0.37\%$ | $53.92\% \pm 0.88\%$ | $46.92\% \pm 0.96\%$ |
| | 0.03 | $\mathbf{86.93\% \pm 0.61\%}$ | $54.49\% \pm 0.56\%$ | $46.45\% \pm 0.33\%$ |
| | 0.05 | $85.99\% \pm 1.30\%$ | $54.33\% \pm 0.33\%$ | $\mathbf{47.01\% \pm 0.36\%}$ |
| | 0.07 | $86.88\% \pm 0.48\%$ | $\mathbf{54.83\% \pm 0.29\%}$ | $46.55\% \pm 0.59\%$ |
| sSELU (Ours) | 0.01 | $87.18\% \pm 0.38\%$ | $\mathbf{57.21\% \pm 0.50\%}$ | $46.26\% \pm 0.49\%$ |
| | 0.017 | $\mathbf{87.29\% \pm 0.20\%}$ | $56.30\% \pm 0.79\%$ | $46.67\% \pm 0.98\%$ |
| | 0.03 | $87.25\% \pm 0.26\%$ | $56.15\% \pm 0.70\%$ | $46.73\% \pm 0.46\%$ |
| | 0.05 | $86.09\% \pm 1.14\%$ | $55.43\% \pm 0.85\%$ | $\mathbf{46.82\% \pm 0.67\%}$ |
| | 0.07 | $86.99\% \pm 0.33\%$ | $55.06\% \pm 0.39\%$ | $46.69\% \pm 0.64\%$ |
| lSELU (Ours) | 0.01 | $83.19\% \pm 0.23\%$ | $54.61\% \pm 0.82\%$ | $44.77\% \pm 0.38\%$ |
| | 0.017 | $85.80\% \pm 0.28\%$ | $55.51\% \pm 0.99\%$ | $46.46\% \pm 0.66\%$ |
| | 0.03 | $\mathbf{86.69\% \pm 0.84\%}$ | $\mathbf{56.78\% \pm 0.44\%}$ | $\mathbf{47.80\% \pm 0.58\%}$ |
| | 0.05 | $86.15\% \pm 1.29\%$ | $56.35\% \pm 0.71\%$ | $47.71\% \pm 0.34\%$ |
| | 0.07 | $86.64\% \pm 0.47\%$ | $54.47\% \pm 1.59\%$ | $47.27\% \pm 0.45\%$ |

First of all, under most of $\epsilon$, our lSELU and sSELU are comparable or even better than dSELU. In particular, sSELU achieves consistent accuracy improvement on CIFAR-10 and CIFAR-100, while lSELU has better performance on Tiny ImageNet. Second, the results show that $\epsilon \approx 1/L \approx 0.017$ is not always the best choice for dSELU, lSELU, and sSELU, but the best accuracy achieved in our sSELU and lSELU are under relatively smaller $\epsilon$ than dSELU (Chen et al., 2020b). These two observations accord with our arguments in Section 4 on the selection of proper $\epsilon$.

## 6.3 LARGE SCALE SNN

In this part, we evaluate our conclusions in Section 5. First, adding Weight Centralization or Mixup successfully solve the gradient explosion problem in SELU, lSELU, and sSELU. Second, in dSELU, lSELU, and sSELU, making $\lambda$ trainable brings additional performance improvement than using Mixup alone. Moreover, after relaxing the constraint $\lambda \geq 1$ to $\lambda \geq 0.5$, the test accuracy "lSELU ($\lambda \geq 0.5$)+Mixup" drops by 8.25%, this demonstrates the importance of constraining $\lambda$ to be no less than 1. Last but not least, by combining centralized lSELU and sSELU with Mixup and trainable $\lambda$, we achieve 71.82% and 71.95% top-1 accuracy.

Table 2: Test accuracy under different configurations on ImageNet (Cl=95%).

| Method | Test Acc. |
|---|---|
| BN + ReLU | $71.42\% \pm 0.07\%$ |
| BN + ReLU + Mixup | $71.54\% \pm 0.36\%$ |
| SELU(Klambauer et al., 2017) | Explode in the first epoch |
| SELU + Weight Centralization | $69.40\% \pm 0.09\%$ |
| SELU + Mixup | $71.37\% \pm 0.14\%$ |
| dSELU(Chen et al., 2020b) | $69.63\% \pm 0.10\%$ |
| dSELU + Weight Centralization | $69.11\% \pm 0.08\%$ |
| dSELU + Mixup | $71.65\% \pm 0.09\%$ |
| dSELU + Mixup (trainable $\lambda$) | $71.74\% \pm 0.09\%$ |
| lSELU | Explode in the first epoch |
| lSELU + Weight Centralization | $70.33\% \pm 0.05\%$ |
| lSELU + Mixup | $70.48\% \pm 0.06\%$ |
| lSELU + Mixup (trainable $\lambda$) | $71.55\% \pm 0.09\%$ |
| lSELU (central) + Mixup (trainable $\lambda$) | $71.82 \pm 0.09\%$ |
| lSELU ($\lambda \geq 0.5$)+Mixup | $63.30\% \pm 0.25\%$ |
| sSELU | Explode in the first epoch |
| sSELU + Weight Centralization | $69.07\% \pm 0.11\%$ |
| sSELU + Mixup | $71.42\% \pm 0.04\%$ |
| sSELU + Mixup (trainable $\lambda$) | $71.77\% \pm 0.09\%$ |
| sSELU (central) + Mixup (trainable $\lambda$) | $\mathbf{71.95\% \pm 0.07\%}$ |

## 7 CONCLUSION

In this paper, we analyze the forward and backward pass signals in SNNs and redefine the self-normalization property. Two novel activation functions, lSELU and sSELU, are developed under this definition. A constrained optimization program is proposed to solve the optimal configurations. Moreover, we reveal the reason behind the performance degradation of SNN under large fan-in, and several solutions are proposed. With our novel methods, advanced results are achieved on multiple benchmarks. Our study demonstrates a new research direction for the design of activation functions.

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

# A    PROOFS

## A.1    SIGNAL PROPAGATION IN DEEP NEURAL NETWORKS

For convenience, we denote the Jacobian matrix $\frac{\partial f(\boldsymbol{h}_{l-1})}{\boldsymbol{h}_{l-1}}$ as $\boldsymbol{D}_{l-1}$, and $tr(\boldsymbol{W}_l\boldsymbol{W}_l^T)tr(\boldsymbol{D}_{l-1}\boldsymbol{D}_{l-1}^T)$ as $\chi_l$, where $tr$ is the normalized trace.

**Proposition A.1** *(**Forward Signal under Mean Field Theory**) Under the formulation, notations, and assumptions above, the evolution of the second moment of pre-activations $q_{l\in[1,L]}$ in the forward pass can be described with*

$$q_l = q_0\Pi_{i=1}^l \frac{\chi_i}{1+\delta_{q_{i-1}}}, \quad l = 1, ..., L. \tag{14}$$

---

*Proof.* Under Assumption 1 & 2, the pre-activation vector of input in layer $l$ can be characterized with a Gaussian random variable $x = \sqrt{q_l}z$, where $z$ is a random variable following $N(0,1)$. With these definitions, we can investigate how $q$ evolves between layer $l-1$ and $l$:

$$q_l = \frac{1}{N_l}\left(\boldsymbol{W}_l f(\boldsymbol{h}_{l-1})+\boldsymbol{b}_l\right)^T\left(\boldsymbol{W}_l f(\boldsymbol{h}_{l-1})+\boldsymbol{b}_l\right) = \sigma_b^2 + \frac{1}{N_l}f(\boldsymbol{h}_{l-1})^T\boldsymbol{U}^T\boldsymbol{\Lambda}\boldsymbol{U}f(\boldsymbol{h}_{l-1}), \tag{15}$$

where $\sigma_b^2 = \frac{1}{N_l}\boldsymbol{b}_l^T\boldsymbol{b}_l$, $\boldsymbol{U}$ is an Orthogonal matrix, and $\boldsymbol{\Lambda}$ is a diagonal matrix of eigenvalues in $\boldsymbol{W}_l^T\boldsymbol{W}_l$. We characterize the diagonal entries in $\boldsymbol{\Lambda}$ with random variable $\lambda$ whose probability density function is $p(\lambda)$. With Assumption 3, we have

$$q_l = \sigma_b^2 + \frac{N_{l-1}}{N_l}\int f^2(\sqrt{q_{l-1}}z)\frac{e^{-\frac{z^2}{2}}}{\sqrt{2\pi}}dz\int \lambda p(\lambda)d\lambda = \sigma_b^2 + tr(\boldsymbol{W}_l\boldsymbol{W}_l^T)\int f^2(\sqrt{q_{l-1}}z)\frac{e^{-\frac{z^2}{2}}}{\sqrt{2\pi}}dz, \tag{16}$$

Then, we substitute equation 2 into equation 16 which yields

$$q_l = \sigma_b^2 + \frac{q_{l-1}}{1+\delta_{q-1}}tr(\boldsymbol{W}_l\boldsymbol{W}_l^T)\int \left[f'(\sqrt{q_{l-1}}z)\right]^2\frac{e^{-\frac{z^2}{2}}}{\sqrt{2\pi}}dz = \sigma_b^2 + \frac{q_{l-1}}{1+\delta_{q-1}}tr(\boldsymbol{W}_l\boldsymbol{W}_l^T)tr(\boldsymbol{D}_{l-1}\boldsymbol{D}_{l-1}^T). \tag{17}$$

As the bias vector is usually initialized with zero and shared among multiple feature entries, $\sigma_b$ has a lower impact than the second term. Therefore, if we neglect the $\sigma_b^2$, with the notation $\chi_l = tr(\boldsymbol{W}_l\boldsymbol{W}_l^T)tr(\boldsymbol{D}_{l-1}\boldsymbol{D}_{l-1}^T)$, we have

$$q_l = q_0\Pi_{i=1}^l\frac{\chi_i}{1+\delta_{q_{i-1}}}, \quad l = 1, ..., L. \tag{18}$$

**Proposition A.2** *(**Backward Gradient under Block Dynamical Isometry**) Under the formulation, notations, and assumptions above, the evolution of the Frobenius norm of the gradient in the backward pass can be described with*

$$E\left[||\frac{\partial\mathcal{L}}{\partial\boldsymbol{h}_l}||_2^2\right]/E\left[||\frac{\partial\mathcal{L}}{\partial\boldsymbol{h}_0}||_2^2\right] \approx \Pi_{i=1}^l\frac{1}{\chi_i}. \tag{19}$$

---

*Proof.* Given the gradient $\frac{\partial\mathcal{L}}{\partial\boldsymbol{h}_l}$, with the chain rule, we have $\frac{\partial\mathcal{L}}{\partial\boldsymbol{h}_{l-1}} = \boldsymbol{D}_l^T\boldsymbol{W}_l^T\frac{\partial\mathcal{L}}{\partial\boldsymbol{h}_l}$ and

$$\frac{\partial\mathcal{L}}{\partial\boldsymbol{h}_0} = \Pi_{i=1}^l\boldsymbol{D}_i^T\boldsymbol{W}_i^T\frac{\partial\mathcal{L}}{\partial\boldsymbol{h}_l}. \tag{20}$$

In particular, we are interested in the Frobenius norm of $\frac{\partial\mathcal{L}}{\partial\boldsymbol{h}_l}$ represented as $||\frac{\partial\mathcal{L}}{\partial\boldsymbol{h}_l}||_2^2$. According to Chen et al. (2020b), its expectation can be computed with

$$E\left[||\frac{\partial\mathcal{L}}{\partial\boldsymbol{h}_0}||_2^2\right] \approx tr\left(\left(\Pi_{i=1}^l\boldsymbol{D}_i^T\boldsymbol{W}_i^T\right)^T\left(\Pi_{i=1}^l\boldsymbol{D}_i^T\boldsymbol{W}_i^T\right)\right) = E\left[||\frac{\partial\mathcal{L}}{\partial\boldsymbol{h}_l}||_2^2\right]. \tag{21}$$

Chen et al. (2020b) proves the theorem as follows.

**Definition 3** *Chen et al. (2020b) (* $k^{th}$ ***Moment Unitarily Invariant) Let*** $\{\mathbf{A_i}\} := \{\mathbf{A_1}, \mathbf{A_2}..., \mathbf{A_L}\}$ *be a series independent random matrices. Let* $\{\mathbf{U_i}\} := \{\mathbf{U_1}, \mathbf{U_3}..., \mathbf{U_L}\}$ *be a series independent haar unitary matrices independent of* $\{\mathbf{A_1}, \mathbf{A_2}..., \mathbf{A_L}\}$. *We say that* $(\Pi_i \mathbf{A_i})(\Pi_i \mathbf{A_i})^T$ *is the* $k^{th}$ *moment unitarily invariant if* $\forall 0 < p \le k$, *we have*

$$tr\left(\left((\Pi_i\mathbf{A_i})(\Pi_i\mathbf{A_i})^T\right)^p\right) = tr\left(\left((\Pi_i\mathbf{U_i}\mathbf{A_i})(\Pi_i\mathbf{U_i}\mathbf{A_i})^T\right)^p\right). \tag{22}$$

**Theorem A.1** *Chen et al. (2020b) (**Multiplication**). Given* $\mathbf{J} := \Pi_{i=L}^1 \mathbf{J_i}$, *where* $\{\mathbf{J_i} \in \mathbb{R}^{m_i \times m_{i-1}}\}$ *is a series of independent random matrices. If* $(\Pi_{i=L}^1 \mathbf{J_i})(\Pi_{i=L}^1 \mathbf{J_i})^T$ *is at least the* $1^{st}$ *moment unitarily invariant (Definition 3), we have*

$$tr\left((\Pi_{i=L}^1\mathbf{J_i})(\Pi_{i=L}^1\mathbf{J_i})^T\right) \Pi_{i=L}^1 tr\left(\mathbf{J_i}\mathbf{J_i}^T\right). \tag{23}$$

Therefore, equation 21 can be further simplified with Theorem A.1 as follows.

$$E\left[||\frac{\partial \mathcal{L}}{\partial \boldsymbol{h}_l}||_2^2\right] / E\left[||\frac{\partial \mathcal{L}}{\partial \boldsymbol{h}_0}||_2^2\right] \approx \Pi_{i=1}^l \frac{1}{tr(\boldsymbol{W}_i\boldsymbol{W}_i^T)tr(\boldsymbol{D}_{i-1}\boldsymbol{D}_{i-1}^T)} = \Pi_{i=1}^l \frac{1}{\chi_i}. \tag{24}$$

## A.2 PROOF OF PROPOSITION 3.1

**Proposition 3.1 (Strength of Self-normalization Property)** Under Definition 2, we represent $\phi(q)$ as a linear interpolation between 1 and $1/q$ as follows.

$$\phi(q) = \begin{cases} 1 + (1 - \gamma_{q<1})(1/q - 1) & if \ q < 1 \\ 1/q + \gamma_{q>1}(1 - 1/q) & if \ q > 1 \end{cases}. \tag{25}$$

where $\gamma_q \in (0, 1)$ is a function of $q$. Then the following conclusions hold:

- The self-normalization property gets stronger when $\gamma_{q<1}$ and $\gamma_{q>1}$ get closer to 0. In particular, $|\gamma_{q<1}| \approx |\gamma_{q>1}| \approx |\frac{d\phi(q)}{dq}|_{q=1} + 1|$ when $q$ is around 1.

- For layer $l$, the gradient explodes under rate $(1 + \delta_{q_l})$, i.e. $\Pi_{i=1}^l (1 + \delta_{q_{i-1}})E\left[||\frac{\partial \mathcal{L}}{\partial \boldsymbol{h}_l}||_2^2\right] = q_0 E\left[||\frac{\partial \mathcal{L}}{\partial \boldsymbol{h}_0}||_2^2\right]$.

*Proof.* When $\gamma_{q<1}$ and $\gamma_{q>1}$ are approaching 0, $\phi(q)$ gets closer to $1/q$. With equation 6, we have $\phi(q) = tr(\boldsymbol{D}_l\boldsymbol{D}_l^T)$ where $\boldsymbol{D}_l$ is the Jacobian matrix of the activation function in layer $l$. As the weights are initialized with $N(0, \frac{1}{N_l})$, we have $tr(\boldsymbol{W}_l\boldsymbol{W}_l^T) = 1$.

In the forward pass, with equation 2, we have $q_{l+1} = \frac{1}{1+\delta_{q_l}}\phi(q_l)q_l$. We can substitute equation 8 and get

$$\begin{cases} 1 - q_{l+1} = \frac{\gamma_{q_l<1}}{1+\delta_{q_l}}(1 - q_l) + (1 - \frac{1}{1+\delta_{q_l}}) & if \ q_l < 1 \\ q_{l+1} - 1 = \frac{\gamma_{q_l>1}}{1+\delta_{q_l}}(q_l - 1) - (1 - \frac{1}{1+\delta_{q_l}}) & if \ q_l > 1 \end{cases}. \tag{26}$$

In the backward pass, with equation 14 and equation 19, we have

$$\frac{E\left[||\frac{\partial \mathcal{L}}{\partial \boldsymbol{h}_l}||_2^2\right]}{q_0 E\left[||\frac{\partial \mathcal{L}}{\partial \boldsymbol{h}_0}||_2^2\right]} = \frac{1/q_l}{\Pi_{i=1}^l(1+\delta_{q_{i-1}})}. \tag{27}$$

Because of

$$E\left[||\frac{\partial \mathcal{L}}{\partial \boldsymbol{h}_{l+1}}||_2^2\right] = \frac{E\left[||\frac{\partial \mathcal{L}}{\partial \boldsymbol{h}_l}||_2^2\right]}{tr(\boldsymbol{D}_l\boldsymbol{D}_l^T)tr(\boldsymbol{W}_{l+1}\boldsymbol{W}_{l+1}^T)} = \frac{E\left[||\frac{\partial \mathcal{L}}{\partial \boldsymbol{h}_l}||_2^2\right]}{\phi(q_l)}, \tag{28}$$

we have

$$
\begin{cases}
\dfrac{E\left[||\frac{\partial \mathcal{L}}{\partial h_{l+1}}||_2^2\right]}{q_0 E\left[||\frac{\partial \mathcal{L}}{\partial h_0}||_2^2\right]} = \dfrac{1}{\Pi_{i=1}^{l}(1+\delta_{q_{i-1}})} \dfrac{1}{q_l+(1-\gamma_{q_l<1})(1-q_l)} = \dfrac{1+\gamma'_{q_l<1}(1/q_l-1)}{\Pi_{i=1}^{l}(1+\delta_{q_{i-1}})}, & if \ q_l < 1 \\[4mm]
\dfrac{E\left[||\frac{\partial \mathcal{L}}{\partial h_{l+1}}||_2^2\right]}{q_0 E\left[||\frac{\partial \mathcal{L}}{\partial h_0}||_2^2\right]} = \dfrac{1}{\Pi_{i=1}^{l}(1+\delta_{q_{i-1}})} \dfrac{1}{1+\gamma_{q_l>1}(q_l-1)} = \dfrac{1/q_l+(1-\gamma'_{q_l>1})(1-1/q_l)}{\Pi_{i=1}^{l}(1+\delta_{q_{i-1}})}, & if \ q_l > 1
\end{cases}, \quad (29)
$$

where

$$
\begin{cases}
\gamma'_{q_l<1} = \dfrac{\gamma_{q_l<1}q_l}{q_l+(1-\gamma_{q_l<1})(1-q_l)} \in (0,1) \\[3mm]
\gamma'_{q_l>1} = \dfrac{\gamma_{q_l>1}q_l}{1+\gamma_{q_l>1}(q_l-1)} \in (0,1)
\end{cases} \quad (30)
$$

are the monotonically increasing functions of $\gamma_{q_l<1}$ and $\gamma_{q_l>1}$, respectively.

Similarly, we can derive how the deviation from the fixed point evolves during the back propagation.

$$
\begin{cases}
\dfrac{E\left[||\frac{\partial \mathcal{L}}{\partial h_{l+1}}||_2^2\right]}{q_0 E\left[||\frac{\partial \mathcal{L}}{\partial h_0}||_2^2\right]} - 1 = \gamma'_{q_l<1}\left(\dfrac{E\left[||\frac{\partial \mathcal{L}}{\partial h_l}||_2^2\right]}{q_0 E\left[||\frac{\partial \mathcal{L}}{\partial h_0}||_2^2\right]} - 1\right) + (1-\gamma'_{q_l<1})(\frac{1}{\Pi_{i=1}^{l}(1+\delta_{q_{i-1}})}-1), & if \ q_l < 1 \\[4mm]
1 - \dfrac{E\left[||\frac{\partial \mathcal{L}}{\partial h_{l+1}}||_2^2\right]}{q_0 E\left[||\frac{\partial \mathcal{L}}{\partial h_0}||_2^2\right]} = \gamma'_{q_l>1}\left(1 - \dfrac{E\left[||\frac{\partial \mathcal{L}}{\partial h_l}||_2^2\right]}{q_0 E\left[||\frac{\partial \mathcal{L}}{\partial h_0}||_2^2\right]}\right) + (1-\gamma'_{q_l>1})(1 - \frac{1}{\Pi_{i=1}^{l}(1+\delta_{q_{i-1}})}), & if \ q_l > 1
\end{cases}. \quad (31)
$$

First of all, when $\delta_{q_i}$ are neglectable, equation 26 and equation 31 can be simplified as

$$
\begin{cases}
1 - q_{l+1} = \gamma_{q_l<1}(1-q_l), \quad \dfrac{E\left[||\frac{\partial \mathcal{L}}{\partial h_{l+1}}||_2^2\right]}{q_0 E\left[||\frac{\partial \mathcal{L}}{\partial h_0}||_2^2\right]} - 1 = \gamma'_{q_l<1}\left(\dfrac{E\left[||\frac{\partial \mathcal{L}}{\partial h_l}||_2^2\right]}{q_0 E\left[||\frac{\partial \mathcal{L}}{\partial h_0}||_2^2\right]} - 1\right), & if \ q_l < 1 \\[4mm]
q_{l+1} - 1 = \gamma_{q_l>1}(q_l-1), \quad 1 - \dfrac{E\left[||\frac{\partial \mathcal{L}}{\partial h_{l+1}}||_2^2\right]}{q_0 E\left[||\frac{\partial \mathcal{L}}{\partial h_0}||_2^2\right]} = \gamma'_{q_l>1}\left(1 - \dfrac{E\left[||\frac{\partial \mathcal{L}}{\partial h_l}||_2^2\right]}{q_0 E\left[||\frac{\partial \mathcal{L}}{\partial h_0}||_2^2\right]}\right), & if \ q_l > 1
\end{cases}. \quad (32)
$$

As $\gamma'_{q_l<1}$ and $\gamma'_{q_l>1}$ are the monotonically increasing functions of $\gamma_{q_l<1}$ and $\gamma_{q_l>1}$, it is obvious that with smaller $\gamma_{q_l<1}$ and $\gamma_{q_l>1}$, the deviation from the fixed point in both forward and backward passes shrinks faster.

In particular, when $q$ is around the fixed point $q = 1$ as ensured by the second term in equation 7, we can approximate $\phi(q)$ and $1/q$ with their first-order Taylor expansion around $q = 1$, with the definition of $\gamma_{q_l<1}$ and $\gamma_{q_l>1}$ in equation 8, we have

$$
\begin{cases}
\gamma'_{q<1} \approx \gamma_{q<1} = \dfrac{1/(1+\Delta q)-\phi(1+\Delta q)}{1/(1+\Delta q)-1} \approx 1 + \dfrac{d\phi(q)}{dq}\Big|_{q=1} + \dfrac{\epsilon}{\Delta q}, & if \ \Delta q < 0 \\[4mm]
\gamma'_{q>1} \approx \gamma_{q>1} = \dfrac{\phi(1+\Delta q)-1/(1+\Delta q)}{1/(1-1+\Delta q)} \approx 1 + \dfrac{d\phi(q)}{dq}\Big|_{q=1} + \dfrac{\epsilon}{\Delta q}, & if \ \Delta q > 0
\end{cases}. \quad (33)
$$

As a result, we can reduce the number of layers required to diminish the deviation by minimizing $|\frac{d\phi(q)}{dq}|_{q=1} + 1|$.

Then, we discuss the influence of $\delta_q$. The fixed point of the two recursive functions in equation 26 and equation 31 can be computed as

$$
\begin{cases}
1 - q = \dfrac{\delta_{q_l<1}}{1+\delta_{q_l<1}-\gamma_{q<1}}, \quad \dfrac{E\left[||\frac{\partial \mathcal{L}}{\partial h_l}||_2^2\right]}{q_0 E\left[||\frac{\partial \mathcal{L}}{\partial h_0}||_2^2\right]} - 1 = \dfrac{1}{\Pi_{i=1}^{l}(1+\delta_{q_{i-1}})} - 1, & if \ q < 1 \\[4mm]
q - 1 = \dfrac{-\delta_{q_l<1}}{1+\delta_{q_l<1}-\gamma_{q>1}}, \quad 1 - \dfrac{E\left[||\frac{\partial \mathcal{L}}{\partial h_l}||_2^2\right]}{q_0 E\left[||\frac{\partial \mathcal{L}}{\partial h_0}||_2^2\right]} = 1 - \dfrac{1}{\Pi_{i=1}^{l}(1+\delta_{q_{i-1}})}, & if \ q > 1
\end{cases}. \quad (34)
$$

While the fixed point of deviation slightly deviates from 0, in the backward pass, we have $\Pi_{i=1}^{l}(1+\delta_{q_{i-1}})E\left[||\frac{\partial \mathcal{L}}{\partial h_l}||_2^2\right] = q_0 E\left[||\frac{\partial \mathcal{L}}{\partial h_0}||_2^2\right]$, which suggests that the gradient explodes with rate $(1+\delta_{q_l})$ at layer $l$.

## A.3 Proof of Proposition 3.2

**Proposition 3.2 (Normalization of Mean)** Under the assumption that the entries in the weight matrix $w_{ij}$ are independent with the input activations, and their expectation has an upper bound $\mu$,

i.e. $\forall\, i, j, \; E[w_{ij}] \leq \mu$. Then we say multiplication with the weight matrix normalizes the mean if $\mu < \frac{1}{N_{l-1}}$ holds, where $N_{l-1}$ is the fan-in of the current layer $l$. Moreover, the mean is scaled down by ratio smaller than $\mu N_{l-1}$.

---

*Proof.* With equation 1, the $j^{th}$ entry in the output pre-activation $\boldsymbol{h}_l$ can be computed with

$$\boldsymbol{h}_{l,j} = \sum_{i=1}^{N_{l-1}} \boldsymbol{w}_{j,i}\boldsymbol{x}_{l-1,i}. \tag{35}$$

Therefore, with the assumption on independence between weight and input activations, we have

$$E[\boldsymbol{h}_{l,j}] = \sum_{i=1}^{N_{l-1}} E[\boldsymbol{w}_{j,i}\boldsymbol{x}_{l-1,i}] \leq N_{l-1}\mu E\left[\frac{1}{N_{l-1}}\boldsymbol{1}^T\boldsymbol{x}_{l-1}\right]. \tag{36}$$

As the term $E\left[\frac{1}{N_{l-1}}\boldsymbol{1}^T\boldsymbol{x}_{l-1}\right]$ can be viewed as the mean of the input activations, when $\mu < \frac{1}{N_l}$, equation 36 reveals that the mean is reduced after multiplying with the weight matrix.

### A.4 BENEFITS OF HAVING $\lambda \geq 1$

First of all, we show that having $\lambda \approx 1$ helps to maintain the mean of the output activations around 0. As we normalize the mean by multiplying with the weights, we don't require the $E[f(x)] = 0$ when $x \sim N(0,1)$ like Klambauer et al. (2017). However, according to Proposition 3.2, the speed of the mean converging to 0 gets slower when the expectation of entries in the weight matrix deviates from 0 and when the fan-in gets larger. Therefore, it's still ideal to avoid shifting the mean too much when the activations flowing through the activation functions. As shown in Figure 5, we simulate the forward pass in a 64-layer fully-connected neural network, and plot the distribution of output activations in layer 1, 4, 16, and 64.

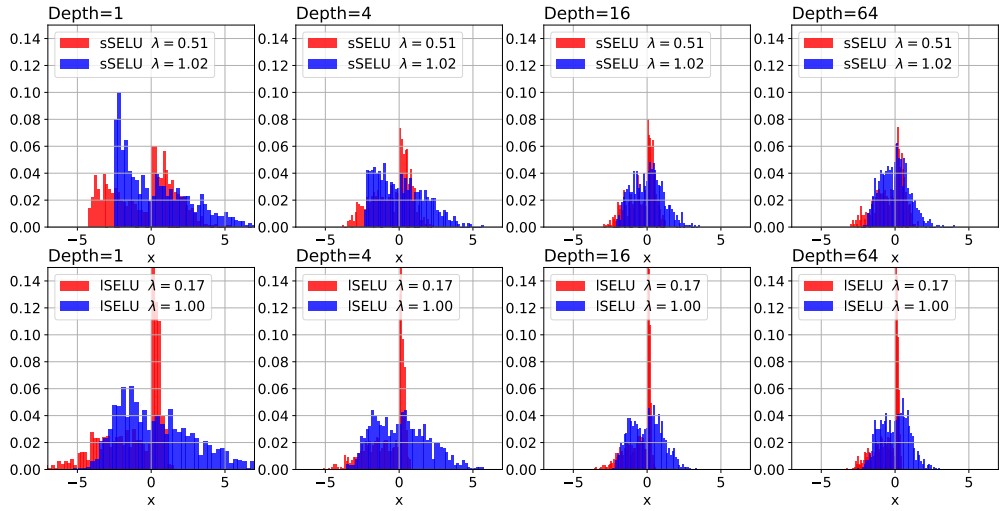

Figure 5: The distribution of output activations of sSELU and lSELU in different layers.

It is obvious that when $\lambda < 1$, a spike around 0 is observed for both sSELU and lSELU, and this leads to a large negative mean of the output activations. On the other hand, for instance, by solving

$$\phi(1) = 1 + \epsilon, \quad \int_{-\infty}^{\infty} f^2(z)\frac{e^{-\frac{z^2}{2}}}{\sqrt{2\pi}}dz = 1, \quad \int_{-\infty}^{\infty} f(z)\frac{e^{-\frac{z^2}{2}}}{\sqrt{2\pi}}dz = 0 \tag{37}$$

under $\epsilon = 0.03$, we have $\lambda \approx 1.0360$ and $1.0362$ for sSELU and lSELU, respectively. Therefore, $\lambda = 1$ is a good starting point for the optimization.

Second, we show that having larger $\lambda$ slows down the gradient explosion in the backward pass. According to the second conclusion in Proposition 3.1, the backward gradient explodes under rate $(1 + \delta_q)$, thus keeping $\delta_q$ low is critical for avoiding gradient explosion. According to the definition in equation 2, $(1 + \delta_q)$ can be computed with

$$1 + \delta_q = \frac{\phi(q)}{q \int_{-\infty}^{\infty} f^2(z) \frac{e^{-\frac{z^2}{2}}}{\sqrt{2\pi}} dz}. \tag{38}$$

We plot the relationship between the maximum $(1 + \delta_q)$ under $q \in (0, 2]$ and $\lambda$ in Figure 6. Obviously, the maximum $(1 + \delta_q)$ decreases when $\lambda$ gets larger. This observation is quite intuitive. The $1 + \delta_i$ characterizes the relative deviation between $E[f^2(x)]$ and $E[(df(x)/dx)^2]E[x^2]$. For the positive pre-activations, we have

Figure 6: Influence of $\lambda$ on $1 + \delta_q$

$$E[f^2(x_+)] = \int_0^{\infty} \lambda \sqrt{q} z \frac{e^{-\frac{z^2}{2}}}{\sqrt{2\pi}} dz = \lambda \int_0^{\infty} \sqrt{q} z \frac{e^{-\frac{z^2}{2}}}{\sqrt{2\pi}} dz = E[(df(x_+)/dx_+)^2]E[x_+^2]. \tag{39}$$

Hence, the deviation is contributed only by the negative part. With a larger $\lambda$, the positive activations are scaled up, thus the negative activations have to be scaled down to preserve the overall second moment. Therefore, the negative part contributes less to the overall second moment, and the relative deviation between $E[f^2(x)]$ and $E[(df(x)/dx)^2]E[x^2]$ gets smaller. All in all, a larger $\lambda$ leads to smaller $\delta_q$, and a smaller $\delta_q$ reduces the gradient explosion rate $(1 + \delta_q)$.

## A.5 THE $\mu N_{l-1}$ WITH INCREASING FAN-IN

Here we empirically illustrate that $\mu N_{l-1}$ increases when the fan-in $N_{l-1}$ gets larger. The experiment is performed on a 32-layer CNN activated with dSELU. We collect the $\mu N_{l-1}$ of each convolutional layer and the final fully-connected layer after each epoch among total 10 epochs. The learning rate is set to 0.005. Let the number of input channels of layer $l$ be $c_l$, the $k$ in each title dSELU$\times k$ indicates that the number of input channel is scaled to $k \times c_l$.

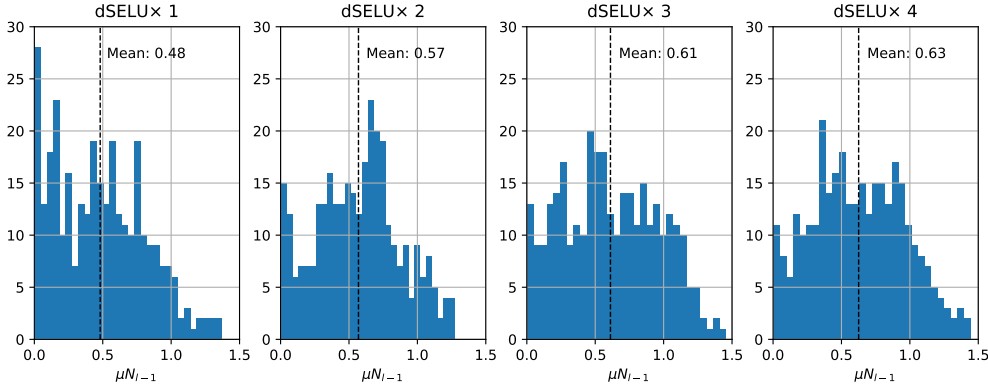

Figure 7: The $\mu N_{l-1}$ with Increasing Fan-in.

As shown in Figure 7, with larger fan-in, the layers tend to have larger $\mu N_{l-1}$. According to Proposition 3.2, larger $\mu N_{l-1}$ will lead to weaker self-normalization property on the mean. Notably, if we further scaling the number of input channels with $k$ greater than 4, gradient explosion happens. These two observations justify our conclusion that the shift of mean is more influential in networks with larger fan-in.

## B EFFICIENT IMPLEMENTATION OF lSELU AND sSELU ON GPU

In this section, we present an efficient implementation for lSELU and sSELU on GPU. In particular, we take lSELU as an example, as the same strategy can be directly applied to sSELU.

---

**Algorithm 1:** Forward Kernel of lSELU.

---

**Data:** Input Feature: $\boldsymbol{X} \in \mathbb{R}^N$, Output Feature: $\boldsymbol{Y} \in \mathbb{R}^N$, $\lambda, \alpha, \beta \in \mathbb{R}$; ThreadIdx: $t$;
  BlockIdx: $b$; Thread Block Size: $T$; Number of Thread Blocks: $B$;

1 **begin**
2    **for** $i = b \times T + t$ **to** $N$ **step** $B \times T$ **do**
3      **if** $\boldsymbol{X}[i] > 0$ **then**
4        $\boldsymbol{Y}[i] = \lambda \times \boldsymbol{X}[i]$
5      **else**
6        $\boldsymbol{Y}[i] = \lambda \times (\alpha \times e^{\boldsymbol{X}[i]} + \beta \times \boldsymbol{X}[i] - \alpha)$

---

**Forward Pass**. The forward pass kernel for lSELU is shown in Algorithm 1. In the forward pass, we have

$$y = \lambda \begin{cases} 1 & if \ x > 0 \\ \alpha e^x + \beta x - \alpha & if \ x \le 0 \end{cases} . \tag{40}$$

The implementation is quite straightforward, all the threads stride across the feature map and element-wisely compute the output activations. The input and output feature maps are treated as 1D array, therefore the kernel achieves good coalescing in both read and write. While we take $T = 1024$, the number of thread blocks $B$ is computed by $B = \lfloor (N + T - 1)/T \rfloor$, so the number of thread blocks is large enough to achieve a high utilization rate.

---

**Algorithm 2:** Backward Kernel of lSELU.

---

**Data:** Gradient of Input Feature: $\boldsymbol{G_X} \in \mathbb{R}^N$, Gradient of Output Feature: $\boldsymbol{G_Y} \in \mathbb{R}^N$; Input
  Feature: $\boldsymbol{X}$; $\lambda, \alpha, \beta \in \mathbb{R}$; Gradient of $\lambda$: $G_\lambda \in \mathbb{R}$; ThreadIdx: $t$; BlockIdx: $b$; Thread
  Block Size: $T$; Number of Thread Blocks: $B$;

1 **begin**
2    float $p = 0$, float $c = 0$
3    **for** $i = b \times T + t$ **to** $N$ **step** $B \times T$ **do**
4      float $x = \boldsymbol{X}[i]$, float $\frac{dy}{dx}$, float $y$, float $g_y = \boldsymbol{G_y}[i]$
5      **if** $\boldsymbol{X}[i] > 0$ **then**
6        $\frac{dy}{dx} = \lambda, y = g_y \times x - c$
7      **else**
8        $\frac{dy}{dx} = \lambda \times (\alpha \times e^x + \beta), y = g_y \times (\alpha \times e^x + \beta \times x - \alpha) - c$
9      float $t = p + y$, $c = (t - p) - y$, $p = t$
10      $\boldsymbol{G_X}[i] = g_y \times \frac{dy}{dx}$
11    $\_\_$syncthreads()
12    $sum$ = BlockReduce($p$)
13    **if** $t = 0$ **then**
14      $atomicAdd(\&G_\lambda, sum)$

---

**Backward Pass**. The backward pass kernel is shown in Algorithm 2. When the $\lambda$ is trainable, the backward pass of lSELU is shown as follows:

$$\frac{\partial \mathcal{L}}{\partial x} = \frac{\partial L}{\partial y}\frac{\partial y}{\partial x} = \frac{\partial L}{\partial y} \times \lambda \begin{cases} 1 & if \ x > 0 \\ \alpha e^x + \beta & if \ x \le 0 \end{cases}, \quad \frac{\partial \mathcal{L}}{\partial \lambda} = \sum \frac{y}{\lambda}\frac{\partial \mathcal{L}}{\partial y}. \tag{41}$$

As the $\frac{\partial \mathcal{L}}{\partial y}$ is used to compute both $\frac{\partial \mathcal{L}}{\partial x}$ and $\frac{\partial \mathcal{L}}{\partial \lambda}$, it can be cached in registers for data reuse (line 4). When the threads stride through the whole feature map, each thread holds a partial sum in a private register $p$ (line 2). In order to avoid the underflow of floating point accumulation, Khan summation algorithm (Higham, 1993) is applied (line 9). At last, we use the block reduction in the CUB library to get the partial sum of the whole thread block, and the final result is atomically added to the $\frac{\partial \mathcal{L}}{\partial \lambda}$.

Different from the forward pass, we choose $B$ to be a few thousands (usually much smaller than $B = \lfloor (N + T - 1)/T \rfloor$). The motivation behind this is that while it is large enough to keep all the

streaming multiprocessors busy, it is also small enough to keep most of the reduction on chip and reduce the atomic transactions.

We evaluate our new CUDA kernels on NVIDIA V100 GPU. We randomly generate an input feature map with size of $[512, 64, 56, 56]$ as the input of the activation function, and we compare the kernel latency of forward and backward passes with the native SELU in PyTorch. The results are summarized in the table below:

Table 3: Forward and Backward pass Latency of SELU and lSELU.

|  | Forward Pass | Backward Pass |
|---|---|---|
| SELU (Klambauer et al., 2017) | 10.487 ms | 14.833 ms |
| lSELU | 10.571 ms | 15.284 ms |
| sSELU | 10.570 ms | 15.260 ms |

Compared with the original SELU, our new implementation with the trainable $\lambda$ only increases the latency by around 2%, which is neglectable. The reason behind this is that the latency of activation functions are bounded by the DRAM bandwidth of GPU (Chen et al., 2020b), and the computation units are underutilized. As our CUDA kernels don't introduce additional DRAM access, it has low impact on the latency.

## C EXPERIMENT SETUP FOR MODERATE-SCALE BENCHMARKS

The experiments in Section 6.1 and 6.2 are based on a 56-layer Convolutional Neural Network shown in Table 4. The $H$ and $W$ are 32 for CIFAR-10, CIFAR-100, and 64 for Tiny ImageNet. Following Klambauer et al. (2017); Chen et al. (2020b), the weights in the convolving kernels are initialized with i.i.d. $N(0, \frac{1}{k_h k_w c_{in}})$, where $k_h$ and $k_w$ are the height and width of the filters and $c_{in}$ is the input channels. The models are optimized with SGD with momentum=0.9, weight decay=0.0005.

Table 4: 56-layer Convolutional Neural Network.

| | Out Size | Serial Network |
|---|---|---|
| conv1 | $W \times H$ | $3 \times 3, 16, s1$ |
| block1 | $W \times H$ | $\begin{bmatrix} 3 \times 3, 16, s1 \\ 3 \times 3, 16, s1 \end{bmatrix} \times 8$ |
| ds1 | $\frac{W}{2} \times \frac{H}{2}$ | $\begin{bmatrix} 3 \times 3, 32, s2 \\ 3 \times 3, 32, s1 \end{bmatrix} \times 1$ |
| block2 | $\frac{W}{2} \times \frac{H}{2}$ | $\begin{bmatrix} 3 \times 3, 32, s1 \\ 3 \times 3, 32, s1 \end{bmatrix} \times 8$ |
| ds2 | $\frac{W}{4} \times \frac{H}{4}$ | $\begin{bmatrix} 3 \times 3, 64, s2 \\ 3 \times 3, 64, s1 \end{bmatrix} \times 1$ |
| block3 | $\frac{W}{4} \times \frac{H}{4}$ | $\begin{bmatrix} 3 \times 3, 64, s1 \\ 3 \times 3, 64, s1 \end{bmatrix} \times 8$ |
| | $1 \times 1$ | average pool, fc |

For Section 6.1, we train the model from scratch for 3190 iterations (10 epochs) on CIFAR-10 under the learning rate 0.015. The choice of learning rate is based on the observation that it is large enough to simulate the fierce update of parameters but also small enough to avoid gradient explosion. Following Chen et al. (2020b), we set $\epsilon = 0.017$ for dSELU, lSELU, and sSELU. We collect the second-moment of the output pre-activations of each convolutional layer as well as the Frobenius norm of backward gradient on the convolving kernels in each iteration.

For Section 6.2, as the model has a relatively small fan-in, we directly apply sSELU and lSELU without techniques mentioned in Section 5. Besides, We clip the gradient by "$[-2, 2]$" for all the experiments to increases stability. All the results are averaged among 4 independent runs to reduce fluctuation. For CIFAR-10 and CIFAR-100, the models are trained with batch size 128 for 130 epochs with the initial learning rate set to 0.01, and decayed to 0.001 at epoch 80. For Tiny ImageNet, the models are trained with batch size 64 for 200 epochs. The initial learning rate is set to 0.001, and decayed by 10 at epoch 130, 180.

For Section 6.3, following Chen et al. (2020b), we choose the Conv MobileNet V1 (Howard et al., 2017). The "Conv" indicates that traditional convolutions rather than depthwise separable convolution is used, as the latter one requires more epochs to converge (Zhou et al., 2020). The model is trained for 90 epochs with batch size 512 under leaning rate 0.02 (decayed by $10\times$ at epoch 60 and 75). Following Zhang et al. (2018), the $\gamma$ for interpolation is drawn from Beta distribution $Beta(0.7, 0.7)$. For all the experiments of dSELU, lSELU, and sSELU, we follow Chen et al. (2020b) and set $\epsilon = 0.06$.

## D   COMPARISON OF PARAMETERS BETWEEN DIFFERENT ACTIVATION FUNCTIONS

We summarize the value of the parameters $\lambda$, $\alpha$, $\beta$, and the corresponding $\gamma_q$ under different configurations in Table 5. According to Proposition 3.1, smaller $\gamma_{q=1}$ will lead to stronger self-normalization property. As shown in Table 5, under the same $\epsilon$, our sSELU and lSELU have lower $\gamma_{q=1}$ compared with dSELU, this justifies our intuition that lSELU and sSELU can be configured to have stronger self-normalization property. Second, the result shows that for each activation function, $\gamma_{q=1}$ increases when $\epsilon$ gets larger. However, as shown in Figure 2, larger $\epsilon$ also leads to larger $\delta_q$, which increases the speed of gradient explosion in the backward pass. Last but not least, for experiments on MobileNet V1, our lSELU and sSELU under $\epsilon = 0.06$ achieve approximately the same $\gamma_{q=1}$ with SELU with $\epsilon \approx 0.0716$, whereas the latter one has a higher gradient explosion rate.

Table 5: $\lambda$, $\alpha$, $\beta$, and the corresponding $\gamma_q$ under different configurations.

| Method | $\epsilon$ | $\lambda$ | $\alpha$ | $\beta$ | $\gamma_{q=1}$ |
|---|---|---|---|---|---|
| 56-layer CNN | | | | | |
| SELU(Klambauer et al., 2017) | | 1.05 | 1.67 | | 0.806 |
| dSELU(Chen et al., 2020b) | 0.01 | 1.37 | 0.48 | | 0.973 |
| | 0.017 | 1.34 | 0.64 | | 0.954 |
| | 0.03 | 1.27 | 0.89 | | 0.919 |
| | 0.05 | 1.17 | 1.25 | | 0.865 |
| | 0.07 | 1.06 | 1.64 | | 0.810 |
| sSELU (Ours) | 0.01 | 1.00 | 4.09 | 0.31 | 0.905 |
| | 0.017 | 1.00 | 3.28 | 0.41 | 0.881 |
| | 0.03 | 1.02 | 2.46 | 0.58 | 0.855 |
| | 0.05 | 1.00 | 2.17 | 0.75 | 0.819 |
| | 0.07 | 1.06 | 1.94 | 0.92 | 0.794 |
| lSELU (Ours) | 0.01 | 1.00 | 0.66 | 0.65 | 0.914 |
| | 0.017 | 1.00 | 0.86 | 0.55 | 0.890 |
| | 0.03 | 1.00 | 1.14 | 0.39 | 0.857 |
| | 0.05 | 1.00 | 1.47 | 0.21 | 0.821 |
| | 0.07 | 1.00 | 1.74 | 0.07 | 0.794 |
| MobileNet V1 | | | | | |
| SELU(Klambauer et al., 2017) | | 1.05 | 1.67 | | 0.806 |
| dSELU(Chen et al., 2020b) | 0.06 | 1.12 | 1.44 | | 0.837 |
| sSELU | 0.06 | 1.00 | 2.04 | 0.84 | 0.805 |
| lSELU | 0.06 | 1.00 | 1.61 | 0.14 | 0.807 |
| sSELU (central) | 0.06 | 1.05 | 1.79 | 0.89 | 0.817 |
| lSELU (central) | 0.06 | 1.05 | 1.54 | 0.08 | 0.818 |

## E   EXPERIMENTS ON FULLY-CONNECTED NEURAL NETWORKS

The performance of SELU proposed in Klambauer et al. (2017) is first demonstrated on fully-connected neural networks. In this section, we compare our sSELU and lSELU against the original SELU in a 64-layer fully-connected neural network on three typical datasets: UCI_miniboone, UCI_adult, and HTRU. The results are summarized in Table 6. As the neural network has 64 layers, we only evaluate sSELU and lSELU at $\epsilon \in \{0.01, 0.017, 0.03\}$. The results show that with all these

three $\epsilon$, our sSELU and lSELU achieve consistent improvement over SELU, which further justifies the effectiveness of our activation functions.

Table 6: Test accuracy on UCI_miniboone, UCI_adult, and HTRU2 (Cl=95%).

| Method | $\epsilon$ | UCI_miniboone | UCI_adult | HTRU2 |
|---|---|---|---|---|
| SELU(Klambauer et al., 2017) | | $92.80\% \pm 0.13\%$ | $84.55\% \pm 0.35\%$ | $97.64\% \pm 0.26\%$ |
| sSELU (Ours) | 0.01 | $\mathbf{93.01\% \pm 0.30\%}$ | $\mathbf{84.68\% \pm 0.23\%}$ | $\mathbf{98.03\% \pm 0.16\%}$ |
| | 0.017 | $92.96\% \pm 0.29\%$ | $84.66\% \pm 0.24\%$ | $97.73\% \pm 0.29\%$ |
| | 0.03 | $92.85\% \pm 0.17\%$ | $84.56\% \pm 0.27\%$ | $97.92\% \pm 0.34\%$ |
| lSELU (Ours) | 0.01 | $\mathbf{93.29\% \pm 0.11\%}$ | $\mathbf{84.86\% \pm 0.18\%}$ | $97.79\% \pm 0.20\%$ |
| | 0.017 | $93.19\% \pm 0.19\%$ | $84.63\% \pm 0.24\%$ | $\mathbf{98.03\% \pm 0.21\%}$ |
| | 0.03 | $92.81\% \pm 0.23\%$ | $84.67\% \pm 0.25\%$ | $98.01\% \pm 0.20\%$ |

