# OpenReview forum: "Redefining The Self-Normalization Property"
_ICLR.cc/2021/Conference — Reject_

### Official Review · AnonReviewer2 · 2020-10-27
**Interesting paper with lacking experiments**

**Rating:** 4
**Confidence:** 4

**Review:**

Summary: The authors propose variants of the SELU activation
function that yield a stronger self-normalization property. The
analysis in done using mean field theory with several
assumptions on the randomness of quantities in a neural
network.

Pros:
a) The paper is clearly written and the line of thought can
easily be followed.
b) The notation is clean.
c) The mathematical formulations are sound.
d) The connections between SNNs and the paper series
by Poole et al. and Schoenholz et al. has been made
quite clear.

Cons:
a) The work has limited relevance due to the
fact that it is unclear whether the proposed property is
indeed crucial for learning and that the empirical results
are biased and far from state-of-the art.
As stated by Goodfellow et al. in their "Deep Learning Book"
(Section 6.3.3), "New hidden unit types that perform roughly comparably to known types are so common
as to be uninteresting." They provide an example using cos() as activation function reaches a test error
below 1% on MNIST. This implies that the machine learning community should be rigorous in the
assessment whether a new activation function is worth publishing in order to avoid drowning literature
about activation functions. Furthermore, the number of potential activation functions for neural networks is
infinite. Therefore, activation function research should be focused around those activation functions with
interesting theoretical properties or -- if no theoretical properties are given -- the new activation
functions should increase the state-of-the-art of predictive performance.
i) The mathematical derivations use several assumptions on the quantities, such as weights and
activations. These assumptions would require commenting on how strong they constrain
the applicability of this theory. The author should comment on how strong those assumptions are
and how they can be leveraged.
ii) It is unclear why a stronger normalization property should improve learning. The authors
should back this more.
iii) Overall, the derivations are rather related to the backward pass. The authors should
include a view on their activation functions in terms of vanishing/exploding gradients (see also [1]).
iv) The main concern of the reviewer are the experiments. Firstly, SNNs were introduced with
relatively large-scale experiments on fully-connected networks. In order to demonstrate
improvements, the suggested activation functions shoudl be compared in that set of experiments.
The  presented experiments are done with architectures that are relatively far
from the current SOTA on CIFAR and Imagenet [2], such that their relevance cannot be judged well.
The authors should introduce their suggested activation functions into SOTA architectures.
v) The presented performance metrics suffer from a hyperparameter selection bias, since
the best epsilon-parameters are selected (section 7.1.). The authors should select
hyperparameters of their method on a validation set.

b) The novely of this could be stated clearer. Large parts of the derivations are
similar to those by Poole et al, 2016. The authors should make their theoretical contributions
clearer.


Questions:
Questions are implicitly contained in the suggestions above.
- Can you elaborate more on the connection to mixup?
- Skip/shortcut connections typically pose a problem both theoretically and practically [3]. How are they handled in this work?



Minor:
1) English editing might be required. Should it say "Redefining THE self-normlization property"?


References:
[1] Hoedt, P.J., Hochreiter, S. and Klambauer, G. Characterising activation functions by their backward dynamics around forward fixed points. Critiquing and Correcting Trends in Machine Learning workshop at NeurIPS 2018.
[2] Xie, Q., Luong, M. T., Hovy, E., & Le, Q. V. (2020). Self-training with noisy student improves imagenet classification. In Proceedings of the IEEE/CVF Conference on Computer Vision and Pattern Recognition (pp. 10687-10698).
[3] Huang, Z., Ng, T., Liu, L., Mason, H., Zhuang, X., & Liu, D. (2020, May). SNDCNN: Self-normalizing deep CNNs with scaled exponential linear units for speech recognition. In ICASSP 2020-2020 IEEE International Conference on Acoustics, Speech and Signal Processing (ICASSP) (pp. 6854-6858). IEEE.

---

> ### Author Response · Authors · 2020-11-24
> **Response to Reviewer 2 (Part 1)**
>
> We greatly appreciate your insightful comments. We have tried our best to revise our paper. The detailed responses are summarized as follows.
>
> **(1) The work has limited relevance due to the fact that it is unclear whether the proposed property is indeed crucial for learning and that the empirical results are biased and far from state-of-the art. As stated by Goodfellow et al. in their "Deep Learning Book" (Section 6.3.3), "New hidden unit types that perform roughly comparably to known types are so common as to be uninteresting." They provide an example using cos() as activation function reaches a test error below 1% on MNIST. This implies that the machine learning community should be rigorous in the assessment whether a new activation function is worth publishing in order to avoid drowning literature about activation functions. Furthermore, the number of potential activation functions for neural networks is infinite. Therefore, activation function research should be focused around those activation functions with interesting theoretical properties or -- if no theoretical properties are given -- the new activation functions should increase the state-of-the-art of predictive performance.**
>
> **Response**. Thank you for your insightful comments. Actually, our paper is indeed focusing on one interesting theoretical property: the self-normalization property. And instead of focusing on a specific activation function, we formulate a definition (Definition 2) that helps identify a class of activation functions with such property. The sSELU and lSELU are only demos to show the effectiveness. In future work, the AutoML techniques can be used, and our theorems can be used to quickly exam the candidate activation functions without actually running the experiments.
>
> The intuition behind the self-normalizing neural network is to encoding the normalization implicitly into the activation functions, such that it can be used when BN is not desirable (e.g. under micro batch size, low bit width).
>
> **(2) The mathematical derivations use several assumptions on the quantities, such as weights and activations. These assumptions would require commenting on how strong they constrain the applicability of this theory. The author should comment on how strong those assumptions are and how they can be leveraged.**
>
> **Response**. The Assumption 1~3 are widely used in existing literatures. In the revised paper, we remove the original assumption in Eq. 4 and replace it with $(1+\delta_q) E[f^2(x)] = E[(df(x)/dx)^2]E[x^2]$, where $x\sim N(0, q)$ (Eq. 2 of the revised paper), and all the new derivations are based on this exact equation.
>
> Beside, to show that they are effective in real applications, in the Section 6.1 of the revised paper, we plot the distribution of the second-moment of forward pre-activation and Frobenius norm of backward gradient on the weights in 10 training epochs under larger learning rate. The result shows that our sSELU and lSELU has flatter and more concentrated distribution in both forward and backward passes compared with SELU and dSELU. This illustrates that our theorems still hold in real application.
>
> **(3) It is unclear why a stronger normalization property should improve learning. The authors should back this more.**
>
> **Response**. The effect of self-normalization property is exactly the same with the normalization achieved by batch normalization. As shown in the Proposition 3.1 of the revised paper, the activation function with stronger nomalization property can fix deviations of statistics in both forward and backward with fewer layer. One extreme situation is batch normalization, which has $\phi(q)=1/q$. Therefore, it could fix the deviation in a single layer. [a]
>
> The faster the deviation can be fixed, the flatter and more concentrated the forward activations and backward gradient will be. As argued in [b], with more or less equal Frobenius Norm of the gradient in diffferent layers (gradient norm equality), the information flow in backward pass can be well preserved. Besides, the gradient explosion / vanishing problems can be better alleviated if the activation function has stronger normalization property.

---

> > ### Author Response · Authors · 2020-11-24
> > **Response to Reviewer 2 (Part 2)**
> >
> > **(4) Overall, the derivations are rather related to the backward pass. The authors should include a view on their activation functions in terms of vanishing/exploding gradients (see also [1]).**
> >
> > **Response**. In the revised paper, we provide detailed derivations on both forward (Proposition A.1) and backward passes (Proposition A.2. ). The derivation on backward pass seems shorter as we leverage several conclusions in [a].  In the proof of Proposition 3.1, we theoretically illustrate the convergense of statistics of both forward activations and backward gradient toward their fixed point.
> >
> > **(5) The main concern of the reviewer are the experiments. Firstly, SNNs were introduced with relatively large-scale experiments on fully-connected networks. In order to demonstrate improvements, the suggested activation function should be compared in that set of experiments. The presented experiments are done with architectures that are relatively far from the current SOTA on CIFAR and Imagenet [2], such that their relevance cannot be judged well. The authors should introduce their suggested activation functions into SOTA architectures.**
> >
> > **Response**: In the Appendix E of the revised paper, we further compare our activation functions against SELU on a 64-layer fully-connected neural network on three datasets used in the original SNN paper: UCI_miniboone, UCI_adult, and HTRU2, and consistent higher accuracy is achieved. Besides, we choose image classification with CNN as the major benchmark is because it is more widely applied.
> >
> > Indeed, the architecture we use are relatively far from the SOTA architectures. However, this is due to two major reasons. First, the SOTA architectures usually have shortcut connections, which is not supported by SNN. According to [a], the networks with shortcut connect has different dynamics with the simple serial CNNs and Fully-connected neural networks, which can be handled with the fixup initialization. However, the fixup initialization cannot be applied to serial networks, and the self-normalizing neural nework fills this vacancy. Second, although these architectures are relatively simple, they are sufficient to demonstrate that our new activation functions deliver stronger self-normalization property.
> >
> > **(6) The presented performance metrics suffer from a hyperparameter selection bias, since the best epsilon-parameters are selected (section 7.1.). The authors should select hyperparameters of their method on a validation set.**
> >
> > **Response**: The hyperparameter selection bias is fixed with following modifications.
> >
> > In Table 1 of the revised paper, we report the performance of all $\epsilon$. Indeed, the best performance is sensitive to $\epsilon$. Nevertheless, arbitrarily taking $\epsilon \approx 1/L$ still achieves consistent performance improvement over SELU [b] and BN. Besides, under the same $\epsilon$, our sSELU and lSELU achieve approximately the same or even better performance than dSELU [a], especially for sSELU on CIFAR-10 and CIFAR-100, lSELU on Tiny ImageNet. These results justify the effectiveness of our new activation functions.
> >
> > In Table 2, we directly use the $\epsilon \approx 1/L$, as it is relatively shallow and $\epsilon \approx 1/16$ is enough to achieve good performance.

---

> > > ### Author Response · Authors · 2020-11-24
> > > **Response to Reviewer 2 (Part 3)**
> > >
> > > **(7) The novely of this could be stated clearer. Large parts of the derivations are similar to those by Poole et al, 2016. The authors should make their theoretical contributions clearer.**
> > >
> > > **Response**: Our theoretical analysis are majorly based on the mean-field theory in Poole et al 2016 [c] and the block dynamical isometry in [a]. However, to the best of our knowledge, we are the first to leverage these theorems on the Self-normalizing neural networks. The major theoretical contributions are summarized as follows.
> > >
> > > * We theoretically illustrate the normalization effectiveness in both forward and backward passes in Proposition 3.1.
> > > * Our new Proposition 3.1 proposes a new value $\gamma_q$ that characterizes the speed that the statistics in forward and backward passes converge to the fixed point, which can be further computed with $\frac{d\phi(q)}{dq}|_{q=1}+1$ when $q$ is around the fixed point $1$.
> > > * We show that adding an additional hyperparameter $\beta$ can improve the self-normalization property without increasing $\epsilon$ or preserve the $\phi(1)=1+\epsilon$, $E[f(x)]=0$ and $E[f^2(x)]=1$ simultaneously (Centralized Activation Function in Sec. 5).
> > > * We theoretically analyze the performance degredation in large-scale SNN and propose several techniques to improved the performance from both the mean-shifting perspective and regularization perspective.
> > >
> > > We have clarified these points in the revised introduction section.
> > >
> > > **(8) Can you elaborate more on the connection to mixup?**
> > >
> > > **Response**: The mixup Augmentation is elaborated in Section 5 of the revised paper. It provides two desirable properties.
> > >
> > > * Mixup reduces the variance / second moment of the inputs, which reduces the gradient explosion rate in the backward pass as well as the deviation of mean caused by the activation functions in the forward pass.
> > > * Mixup provides addition regularization to the training process by constructing new training samples with the linear interpolation between existing ones.
> > >
> > > **(9) Skip/shortcut connections typically pose a problem both theoretically and practically [3]. How are they handled in this work?**
> > >
> > > **Response**: As we mentioned before, the shortcut connections change the dynamics of the network [a] and should be handled by the fixup initialization [a]. Similarly, our work only focus on the network without shortcut connections. However, we believe that our work is a good complementary for the fixup initialization, as the latter one doesn't work one the networks without shortcut connections.
> > >
> > > **(10) English editing might be required. Should it say "Redefining THE self-normalization property"?**
> > >
> > > **Response**: We have corrected it in the revised paper. Thank you for the reminder.
> > >
> > > #### References
> > >
> > > * [a] Z. Chen, L. Deng, B. Wang, G. Li and Y. Xie, "A Comprehensive and Modularized Statistical Framework for Gradient Norm Equality in Deep Neural Networks," in IEEE Transactions on Pattern Analysis and Machine Intelligence, doi: 10.1109/TPAMI.2020.3010201.
> > > * [b] Arpit, Devansh, and Yoshua Bengio. "The benefits of over-parameterization at initialization in deep ReLU networks." *arXiv preprint arXiv:1901.03611* (2019).
> > > * [c] Poole, Ben, et al. "Exponential expressivity in deep neural networks through transient chaos." *Advances in neural information processing systems*. 2016.

---

### Official Review · AnonReviewer3 · 2020-10-28
**Solid effort, but theoretical motivation and experimental evidence aren't convincing enough**

**Rating:** 5
**Confidence:** 3

**Review:**

# Update

I thank the authors for extensive replies and updates to the paper. Most of my questions are answered, and the paper quality is substantially improved. I would not be opposed if other reviewers recommends to accept it. Unfortunately I still can't raise the score and advocate for it myself, since:

1) Table 2 doesn't really show superiority of new nonlinearities, since the bolded (bottom) entry has both trainable $\lambda$, centralization and Mixup, while the baseline of dSELU only has mixup and trainable $\lambda$, no centralization. Without centralization (Mixup + trainable $\lambda$), lSELU/sSELU/dSELU perform comparatively. Without trainable $\lambda$ (only Mixup), they perform a bit worse than dSELU. With only centralization, BN is still better. Further, even after rebuttal, I still believe that making $\lambda$ trainable effectively cancels the preceding theoretical discussion, and makes this subset of results somewhat unrelated to the main idea of the paper. Finally, Table 1 shows a more robust benefit over dSELU on CIFAR-100, but not on CIFAR-10/TinyImageNet (where s/lSELU can be both better and worse than dSELU), which is in my opinion underwhelming given the added implementation complexity.

2) Figure 4 is very promising, but I find that SELU doesn't look that bad on it, which again makes me wonder whether the new improved self-normalization actually matters in Tables 1/2, especially given gradient clipping and other heuristics in Table 2.

So at the moment I find that the proposed nonlinearities are promising in terms of both self-normalization (Figure 4), and generalization (Table 1/2/6), but these results appear to be largely unrelated to each other, and neither of them in separation is strong enough to convince me to try s/lSELU over dSELU (given additional implementation complexity + the boolean hyper-parameter of whether to use lSELU or sSELU) or over BN (given additional hyper-parameter $\epsilon$). I wish the paper either showed clear use-cases where one can't train BN/SELU/dSELU networks at all in reasonable time (but new nonlinearities allowed it due to superior normalization), or more robust generalization results.

Original review below:

# Outline
The paper proposes two new nonlinearities designed to normalize the second moment of activations and the Frobenius norm of gradients, and therefore avoid exploding/vanishing activations/gradients. The nonlinearities are evaluated on multiple image datasets.

# Review

On one hand, I find the theoretical motivation and the idea of adding the new minimization constraint compelling, and empirical performance of nonlinearities promising (notably on CIFAR-100). I appreciate evaluation on multiple datasets, and providing native CUDA code. Further, several decisions along the design process appear well-motivated and backed by experiments (e.g Figure 2, 3, 4).

On the other hand, I find the theoretical contribution to be relatively incremental from Chen et al (2020b), and unrelated to the uptick in generalization performance in Tables 1 and 2 (motivation for deriving the nonlinearities concern improving trainability, which is not studied experimentally in the paper; better generalization is a nice bonus, but is not predicted by theory).

On the empirical side, I believe there are important gaps in the experimental results that leave me uncertain of how robust the improved generalization is, and hence whether new nonlinearities justify the added complexity (I believe the claim that no new hyper-paramaters are introduced is false) over SELU or dSELU from Chen et al (2020b).

As such, at the time of submission neither theory nor experiments appear sufficiently convincing for me to accept the paper, but I am open for discussion.


Below are my specific questions/concerns.


## Motivation/Theory:
1. Do I understand correctly that Definition 2, “new self-normalization definition”, and what follows within section 4 is effectively the same as “partial normalization” in Chen et al (2020b), section 5.3? In either case, I suggest drawing a more explicit connection with Chen et al (2020b) in this section, highlighting precisely what is novel relative to Chen et al (2020b), and potentially softening/clarifying the first bullet point of main theoretical results in the Introduction respectively.
2. One important weakness of this work is that the paper does not evaluate whether the designed nonlinearity ends up doing what it was designed to do. Namely, we know that in finite networks Assumptions 1-3 don’t hold, Eq. (4) is not exact, and, for $\lambda \geq 1$ $\frac{d \phi(q)}{d q}|_{q=1}$ is clearly far from $-1$ in Figure 1, notably even deviating further from it than dSELU in Figure 1.c. As such I am not persuaded that s/lSELU has better gradient behavior than dSELU in practice. One could demonstrate such benefits in different ways, perhaps by showing that s/lSELU allows higher learning rate in deep networks on some toy task, or comparing how the gradient norms evolve with depth in deep random networks compared to dSELU etc. Further, even in generalization experiments, as mentioned in appendix B, both this paper and Chen et al. (2020b) clip the gradients by $[-2; 2]$, which, from my understanding, further conceals whether these nonlinearities help normalize gradients or not.
3. Making $\lambda$ trainable in s/lSELU is a somewhat disappointing decision/necessity since there is no theoretical reason to do so, and from a quick look at the referenced Zhang et al. (2019) I could not find the justification there - could you please clarify which part of the paper you were referencing? Does $\lambda$ remain constrained to $\geq 1$ when trained?
4. I would like the discussion about “large fan-in networks are likely to lose self-normalization on the mean” (section 6, 3rd paragraph) to be either elaborated, referenced, or confirmed empirically (e.g. on a toy task with networks of increasing fan-ins). Precisely, the conclusion is reached under the assumption that the mean of weights $\Delta \mu$ (during training) is independent of fan-in $N_{l-1}$, but isn’t this clearly not true? I.e. weights with larger fan-in are sampled with smaller variance, and I expect them to move less and less during training as the network gets wider, hence increasing fan-in could proportionally decrease $\Delta \mu$. In this case these changes cancel out, and the conclusion does not appear justified.


## Experiments:
1. As presented, experiments do not support the introductory claim that “no additional hyper-parameter is introduced". Firstly, I would prefer if the claim was more explicit, i.e. “no additional hyper-paramater relative to dSELU”. Secondly, even apart from $\epsilon$, there are still hyper-parameters of a) which among s/lSELU to use; b) whether to have $\lambda$ trainable or not; c) whether to use Weight Centering, or mixup, or nothing. From reading the abstract, I was expecting a drop-in, no-hyper-parameter activation, but unfortunately there are quite a few knobs to tune.
2. In both Tables, I believe it’s important to have entries with hyper-parameter-free d/s/lSELU as a baseline, i.e. with $\epsilon = 1 / L$, to get an understanding of what order of improvement is gained from the additional $\epsilon$ hyper-parameter. I’m concerned that If $\epsilon$ is necessary, the relative performance of d/l/sSELU against BN/SELU may be less compelling, especially if the best-performing models were selected on the test set (see next question).
3. In both tables, was a validation set used to select best performing numbers, or were the numbers selected on the test set (please expand appendix B with this detail)?
4. Table 2: what $\epsilon$ was used?
5. In both Tables, it would be useful to see how sensitive different nonlinearities are to $\epsilon$, so I suggest adding respective tables for all values in the appendix (also noting the specific values of $\lambda, \alpha, \beta$).
6. In Table 2: since Chen et al. (2020b) is the primary point of reference for this work, I would argue that for this Table to be convincing, it should also include “dSELU + Mixup” and “dSELU + Mixup + trainable $\lambda$” (and, ideally,  “dSELU + Weight Centralization”).
7. In Table 2, could you please include results with non-trainable $\lambda$ as well?
8. Was trainable $\lambda$ a single shared parameter for the whole network, or one per layer?

## Mentioned references from the paper:
* Chen et al. (2020b): [A Comprehensive and Modularized Statistical Framework for Gradient Norm Equality in Deep Neural Networks](https://arxiv.org/abs/2001.00254)
* Zhang et al. (2019): [Fixup Initialization: Residual Learning Without Normalization]( https://arxiv.org/abs/1901.09321)

---

> ### Author Response · Authors · 2020-11-24
> **Response to Reviewer 3 (Part 1)**
>
> Thanks for your insightful comments. We have carefully revised our paper. The detailed responses are summarized as follows.
>
> **(1) I find the theoretical contribution to be relatively incremental from Chen et al (2020b), and unrelated to the uptick in generalization performance in Tables 1 and 2 (motivation for deriving the nonlinearities concern improving trainability, which is not studied experimentally in the paper; better generalization is a nice bonus, but is not predicted by theory).**
>
> **Response**: In this paper, we still focus on the self-normalization effectiveness rather than generalization, which is similar to Chen et al (2020b) [a]. However,Chen et al (2020b) [a] only coarsely analyzes the backward pass in SELU and provides some insights. In our revised paper, on the basis of Chen et al (2020b)[a]'s conclusion, we have several additional theoretical contributions.
>
> *  While Chen et al (2020b)[a] only focus on the backward pass, we theoretically illustrate the normalization effectiveness in both forward and backward passes in Proposition 3.1.
> * While the Eq. 34 and Eq. 36 in Chen et al (2020b)[a] only give lower/upper bounding, our new Proposition 3.1 proposes a new value $\gamma_q$ that characterizes the speed that the statistics in forward and backward passes converge to the fixed point, which can be further computed with $\frac{d\phi(q)}{dq}|_{q=1}+1$ when $q$ is around the fixed point $1$.
> * While Chen et al (2020b)[a]'s derivation is based on the assumption that SELU is approximately General Linear Transform (Def. 5.1 of [a]), our revised paper takes the difference between $E[f^2(x)]$ and $E[(df(x)/dx)^2]E[x^2]$ into consideration by modeling it with a variable $\delta_q$ (Eq. 2 of the revised paper). We prove that the speed of gradient explosion is actually $(1+\delta_q)$ rather than $(1+\epsilon)$, even though $\delta_{q=1}=\epsilon$. Considering the nonzero $\delta_q$ allows us to provide more insight on the selection of $\epsilon$ (Last paragraph of Section 4) as well as additional theoretical insight behind the Mixup data Augmentation (Section 5).
> * While the dSELU proposed in Chen et al (2020b)[a] still take the form of SELU, we show that adding an additional hyperparameter $\beta$ can improve the self-normalization property without increasing $\epsilon$ or preserve the $\phi(1)=1+\epsilon$, $E[f(x)]=0$ and $E[f^2(x)]=1$ simultaneously (Centralized Activation Function in Sec. 5).
> * We theoretically analyze the performance degredation in large-scale SNN and propose several techniques to improved the performance from both the mean-shifting perspective and regularization perspective.
>
> To support the theoretical conclusions above, we enriched the experiments in the revised paper. Specifically, In Section 6.1, we plot the distribution of the second-moment of forward pre-activation and Frobenius norm of backward gradient under different activation functions. The result shows that our sSELU and lSELU has flatter and more concentrated distribution in both forward and backward passes compared with SELU and dSELU, which implies stronger self-normalization property. In Section 6.2, we use a 56-layer network to better demonstrate the self-normalization property, where as Chen et al (2020b)[a] only takes 32 layers. In Section 6.3, the techniques proposed in Section 5 successfully improve the performance on ImageNet. For example, weight centralization and Mixup enable using SELU on MobileNet V1, and our centralized actiavtion function proposed at the end of Section 5 achieves 0.41% higher accuracy than the BN baseline.

---

> > ### Author Response · Authors · 2020-11-24
> > **Response to Reviewer 3 (Part 2)**
> >
> > **(2) On the empirical side, I believe there are important gaps in the experimental results that leave me uncertain of how robust the improved generalization is, and hence whether new nonlinearities justify the added complexity (I believe the claim that no new hyper-parameters are introduced is false) over SELU or dSELU from Chen et al (2020b).**
> >
> > **Response**: In this paper, we still focus on the self-normalization effectiveness rather than the generalization.
> >
> > In the Section 6.2 of the revised paper, we evaluate the accuracy of CIFAF-10, CIFAR-100, and Tiny ImageNet on a 56-layer CNN, which shows that our sSELU and lSELU could achieve better performance than dSELU and SELU in the literature. We believe this improvement is majorly due to the improved self-normalization property. As the experiment in Section 6.1 shows that the sSELU and lSELU has flatter and more concentrated distribution in both forward and backward passes.
> >
> > In the Section 6.3, we focus on the performance degredation when applying these activation functions to larger networks like MobileNet V1. Theoretically, we show that the degredation is mainly due to the shift of mean and lack of regularization. In Section 6.3, we show that the techniques inspired by the theoretical conclusions greatly improve the performance.
> >
> > For the claim that "no new hyper-parameters are introduced", while the new $\beta$ can be solved with the constrained optimization program in Eq. 11 in the revised paper, the $\epsilon$ can be configured under the same way as dSELU, and we add some additional discussion on choosing $\epsilon$ in the last paragraph of Section 4. The reason that dSELU doesn't tune $\epsilon$ is that they use a relatively shallow network (32 layers), for which $\epsilon \approx 1/L$ is enough to achieve a good trade-off between self-normalization property and speed of gradient explosion. In the Section 6.2 of the revised paper, we show that in the 56-layer CNN, $\epsilon$ of dSELU also has to be tuned to achieve the best accuracy. Although tuning the $\epsilon$ may be less convenient, Table 1 shows that arbitrarily taking $\epsilon \approx 1/L$ still achieves consistent performance improvement over SELU [b]. Besides, under the same $\epsilon$, our sSELU and lSELU achieve approximately the same or even better performance than dSELU [a], especially for sSELU on CIFAR-10 and CIFAR-100, lSELU on Tiny ImageNet. The Mixup Data Augmentation, Weight Centralization, and centralized activation function proposed in Section 5 are three techniques to improve the performance on large models like MobileNet V1 for ImageNet, and the result shows that Mixup + trainable $\lambda$ + centralized activation function consistently achieves the best performance.
> >
> > **(3) Do I understand correctly that Definition 2, “new self-normalization definition”, and what follows within section 4 is effectively the same as “partial normalization” in Chen et al (2020b), section 5.3? In either case, I suggest drawing a more explicit connection with Chen et al (2020b) in this section, highlighting precisely what is novel relative to Chen et al (2020b), and potentially softening/clarifying the first bullet point of main theoretical results in the Introduction respectively.**
> >
> > **Response**: Our theoretical contributions over Chen et al (2020b) are well summarized in the response to your first question. For summary, although Definition 2 is basically a formally defined partial normalization in Chen et al (2020b) [a], our Proposition 3.1 further theoretically defines the strength of the self-normalization property, and discussion in Section 5 reveals the reasons behind the performance degredation in large models. We also clarify these point in the revised Introduction section.

---

> > > ### Author Response · Authors · 2020-11-24
> > > **Response to Reviewer 3 (Part 3)**
> > >
> > > **(4) One important weakness of this work is that the paper does not evaluate whether the designed nonlinearity ends up doing what it was designed to do. Namely, we know that in finite networks Assumptions 1-3 don’t hold, Eq. (4) is not exact, and, for $\lambda \ge 1 \frac{d\phi(q)}{dq}|_{q=1}$is clearly far from −1 in Figure 1, notably even deviating further from it than dSELU in Figure 1.c. As such I am not persuaded that s/lSELU has better gradient behavior than dSELU in practice. One could demonstrate such benefits in different ways, perhaps by showing that s/lSELU allows higher learning rate in deep networks on some toy task, or comparing how the gradient norms evolve with depth in deep random networks compared to dSELU etc. Further, even in generalization experiments, as mentioned in appendix B, both this paper and Chen et al. (2020b) clip the gradients by [−2;2], which, from my understanding, further conceals whether these nonlinearities help normalize gradients or not.**
> > >
> > > **Response**: In the revised paper, we remove the original assumption in Eq. 4 and replace it with $(1+\delta_q) E[f^2(x)] = E[(df(x)/dx)^2]E[x^2]$, where $x\sim N(0, q)$ (Eq. 2 of the revised paper), and all the new derivations are based on this exact equation.
> > >
> > > It is true that for $\lambda \ge 1 \frac{d\phi(q)}{dq}|{q=1}$ is far from -1. However, the $\frac{d\phi(q)}{dq}|{q=1}$ of our sSELU and lSELU is still closer to $-1$ compared with SELU and dSELU under the same $\epsilon$, which implies stronger self-normalization property. In particular, we summarize the $\gamma_{q=1}=\frac{d\phi(q)}{dq}|_{q=1} +1 $ in Table 5 of Appendix D. The curves in Figure 1 that deviating further form dSELU are only used to show that the distance between $\phi(q)$ and $1/q$ can be configured with the parameters, rather than representing the optimal values.
> > >
> > > To show that sSELU and lSELU have better gradient behavior than dSELU, in the Section 6.1 of the revised paper, we plot the distribution of the second-moment of forward pre-activation and Frobenius norm of backward gradient on the weights in 10 training epochs under larger learning rate. The result shows that our sSELU and lSELU has flatter and more concentrated distribution in both forward and backward passes compared with SELU and dSELU. This illustrates that our sSELU and lSELU has stronger self-normalization property.
> > >
> > > According to Figure 4 of the revised paper, the [-2, 2] is a relatively loose clip bound, which is only added to alleviate the influence of some large outliers.
> > >
> > > **(5) Making $\lambda$ trainable in s/lSELU is a somewhat disappointing decision/necessity since there is no theoretical reason to do so, and from a quick look at the referenced Zhang et al. (2019) I could not find the justification there - could you please clarify which part of the paper you were referencing? Does λ remain constrained to ≥1when trained?**
> > >
> > > **Response**: Making $\lambda$ trainable is not targeting on learning a good $\lambda$. Instead, togather with the bias of the convolutional layer, they serve the same function as the affine transform in batch normalization to increase the representation power of the network, which is equivalent with the "scalar multiplier (initialized at 1)" in Fixup initialization (3rd term in the definition, page 5 [b]). We have clarified this point in Sec. 5 Mixup Data Augmentation in the revised paper.
> > >
> > > **(6) I would like the discussion about “large fan-in networks are likely to lose self-normalization on the mean” (section 6, 3rd paragraph) to be either elaborated, referenced, or confirmed empirically (e.g. on a toy task with networks of increasing fan-ins). Precisely, the conclusion is reached under the assumption that the mean of weights Δμ (during training) is independent of fan-in Nl−1, but isn’t this clearly not true? I.e. weights with larger fan-in are sampled with smaller variance, and I expect them to move less and less during training as the network gets wider, hence increasing fan-in could proportionally decrease Δμ. In this case these changes cancel out, and the conclusion does not appear justified.**
> > >
> > > **Response**: It is true that $\mu$ is not independent of fan-in $N_{l-1}$. However, in the Appendix A.5 of the revised paper, we empirically show that the  $\mu$ decreases under a lower speed compared with the increase of $N_{l-1}$, and networks with larger fan-in tend to have larger $\mu N_{l-1}$, which implies weaker self-normalization property on mean.

---

> > > > ### Author Response · Authors · 2020-11-24
> > > > **Response to Reviewer 3 (Part 4)**
> > > >
> > > > **(7) As presented, experiments do not support the introductory claim that “no additional hyper-parameter is introduced". Firstly, I would prefer if the claim was more explicit, i.e. “no additional hyper-paramater relative to dSELU”. Secondly, even apart from ϵ, there are still hyper-parameters of a) which among s/lSELU to use; b) whether to have λ trainable or not; c) whether to use Weight Centering, or mixup, or nothing. From reading the abstract, I was expecting a drop-in, no-hyper-parameter activation, but unfortunately, there are quite a few knobs to tune.**
> > > >
> > > > **Response**. We have clarified in the introduction that it is "no additional hyper-parameter relative to dSELU".
> > > >
> > > > The $\epsilon$ can be configured under the same way as dSELU, and we add some additional discussion on choosing $\epsilon$ in the last paragraph of Section 4. The reason that dSELU doesn't tune $\epsilon$ is that they use a relatively shallow network (32 layers), for which $\epsilon \approx 1/L$ is enough to achieve a good trade-off between self-normalization property and speed of gradient explosion. In the Section 6.2 of the revised paper, we show that in the 56-layer CNN, $\epsilon$ of dSELU also has to be tuned to achieve the best accuracy. Although tuning the $\epsilon$ may be less convenient, Table 1 shows that arbitrarily taking $\epsilon \approx 1/L$ still achieves consistent performance improvement over SELU [b]. Besides, under the same $\epsilon$, our sSELU and lSELU achieve approximately the same or even better performance than dSELU [a], especially for sSELU on CIFAR-10 and CIFAR-100, lSELU on Tiny ImageNet.
> > > >
> > > > The Mixup Data Augmentation, Weight Centralization, and centralized activation function proposed in Section 5 are three techniques to improve the performance on large models like MobileNet V1 for ImageNet, and the result shows that Mixup + trainable $\lambda$ + centralized activation function consistently achieves the best performance. Basically, they can be directly used to improve the performance on large scale models without making decisions, and they don't even conflict with each other.
> > > >
> > > > **(8) In both Tables, I believe it’s important to have entries with hyper-parameter-free d/s/LSELU as a baseline, i.e. with ϵ=1/L, to get an understanding of what order of improvement is gained from the additional ϵ hyper-parameter. I’m concerned that If ϵ is necessary, the relative performance of d/l/sSELU against BN/SELU may be less compelling, especially if the best-performing models were selected on the test set (see next question).**
> > > >
> > > > **Response**: In the revised paper, we report the accuracy under all the 5 $\epsilon$ in Table 1, including $\epsilon = 0.017 \approx 1/56$. The result shows that arbitrarily taking $\epsilon \approx 1/L$ still achieves consistent performance improvement over SELU [b] and BN. Besides, under the same $\epsilon$, our sSELU and lSELU achieve approximately the same or even better performance than dSELU [a], especially for sSELU on CIFAR-10 and CIFAR-100, lSELU on Tiny ImageNet.
> > > >
> > > > In Table 2, we take $\epsilon=0.06\approx 1/16$ following Chen et al (2020b)[a], which is already free of tunable $\epsilon$.
> > > >
> > > > **(9) In both tables, was a validation set used to select best performing numbers, or were the numbers selected on the test set (please expand appendix B with this detail)?**
> > > >
> > > > **Response**. In the original submission, we only report the best accuracy due to the limit of space. In the revised paper, we report the accuracy under all the 5 $\epsilon$ in Table 1.  Besides, the $\alpha=0.7$ for mixup is arbitrarily taken from [a] and [b].
> > > >
> > > > **(10) Table 2: what ϵ was used?**
> > > >
> > > > **Response**: $0.06\approx 1/16$, which is same as Chen et al (2020b)[a]
> > > >
> > > > **(11) In both Tables, it would be useful to see how sensitive different nonlinearities are to ϵ, so I suggest adding respective tables for all values in the appendix (also noting the specific values of λ,α,β).**
> > > >
> > > > **Response**: In Table 1 of the revised paper, we report the performance of all $\epsilon$. Indeed, the best performance is sensitive to $\epsilon$. Nevertheless, arbitrarily taking $\epsilon \approx 1/L$ still achieves consistent performance improvement over SELU [b] and BN. Besides, under the same $\epsilon$, our sSELU and lSELU achieve approximately the same or even better performance than dSELU [a], especially for sSELU on CIFAR-10 and CIFAR-100, lSELU on Tiny ImageNet. These results justify the effectiveness of our new activation functions. In Table 2, we directly use the $\epsilon \approx 1/L$, as it is relatively shallow and $\epsilon \approx 1/16$ is enough to achieve good performance.
> > > >
> > > > The results of $\lambda$, $\alpha$, $\beta$ of different configurations are reported in Appendix D of the revised paper.

---

> > > > > ### Author Response · Authors · 2020-11-24
> > > > > **Response to Reviewer 3 (Part 5)**
> > > > >
> > > > > **(12) In Table 2: since Chen et al. (2020b) is the primary point of reference for this work, I would argue that for this Table to be convincing, it should also include “dSELU + Mixup” and “dSELU + Mixup + trainable λ” (and, ideally, “dSELU + Weight Centralization”).**
> > > > >
> > > > > **Response**: We have included these experiments in the Table 2 of the revised paper. However, using Mixup, Weight Centralization, and trainable $\lambda$ to improve the performance are also our contributions. As the intuition behind these methods are obtained via the theoretical analysis in Section 5.
> > > > >
> > > > > **(13) In Table 2, could you please include results with non-trainable λas well?**
> > > > >
> > > > > **Response**: We have included the non-trainable $\lambda$ in the revised Table 2. The performance drops by a little compared with those with trainable $\lambda$s. The intuition behind this is that the trainable $\lambda$ serves as a scalar multipler that increases the representational power of the neural network
> > > > >
> > > > > **(14) Was trainable λ a single shared parameter for the whole network, or one per layer?**
> > > > >
> > > > > **Response**: It is one per layer. Notably, our model with trainable $\lambda$ still has fewer parameters compared with batch normalization, as the batch normalization of each layer contains a trainable vector $\gamma$ that serves the same purpose.
> > > > >
> > > > > #### References
> > > > >
> > > > > * [a] Z. Chen, L. Deng, B. Wang, G. Li and Y. Xie, "A Comprehensive and Modularized Statistical Framework for Gradient Norm Equality in Deep Neural Networks," in IEEE Transactions on Pattern Analysis and Machine Intelligence, doi: 10.1109/TPAMI.2020.3010201.
> > > > > * [b] Zhang, H., Dauphin, Y. N., & Ma, T. (2019). Fixup initialization: Residual learning without normalization. *arXiv preprint arXiv:1901.09321*.

---

> > > > ### Comment · AnonReviewer3 · 2020-11-24
> > > > **Quick question on trainable $\lambda$**
> > > >
> > > > Thank you for your replies, could you please clarify this part?
> > > >
> > > > > **Does λ remain constrained to ≥1when trained?**
> > > >
> > > > (unless I missed it somewhere in your reply/revision)

---

> > > > > ### Author Response · Authors · 2020-11-24
> > > > > **Trainable $\lambda$**
> > > > >
> > > > > Thank you for your comment.
> > > > >
> > > > > $\lambda$ doesn't remain constrained to $\ge 1$ when trained. Actually, it behaviors just like the $\gamma$ in the affine transform of Batch Normalization. However, it doesn't conflict with our theory.
> > > > >
> > > > > Having $\lambda \ge 1$ is most important when determining the value of $\lambda$, $\alpha$, and $\beta$. Intuitively, this is because we want to avoid generating an activation function with a strange shape and undesirable properties (e.g. $\lambda=0.17$ in Figure 1 d). Theoretically, when we solve these three parameters, we want it to satisfy the three conditions in Eq. 7 in Definition 2. As discussed in Appendix A.4, having $\lambda \ge 1$ helps to avoid the large shift of the mean caused by the activation function as well as reduce the $\delta_q$.
> > > > >
> > > > > On the other hand, during training, when $\lambda < 1$, it doesn't greatly influence the mean of the output activations as all the output activations in the same layer share the same $\lambda$. Moreover, similar to Mixup that reduces the second moment in the forward pass, when $\lambda < 1$, it also decreases the second moment in the forward pass, which reduces $\delta_q$, and the gradient explosion rate $(1+\delta_q)$ also decreases in the backward pass.

---

### Official Review · AnonReviewer1 · 2020-10-28
**Interesting results but needs better experiments**

**Rating:** 5
**Confidence:** 3

**Review:**

In this paper, the authors propose a new definition for the self-normalization property of a network. Using this definition, the authors propose two new activation functions: sSELU and lSELU. Performance of the new activation functions is tested on benchmarks like CNNs on CIFAR10/CIFAR100/Tiny-ImageNet and MobileNet on ImageNet.

The new definition of self-normalization is interesting and could potentially be useful. The paper is also written reasonably clearly (see below for minor comments on how I think the writing could be improved). However, I think the experimental section of the paper using the two proposed activation functions have a number of limitations (which I describe below) that explain my rating. I am happy to raise my score if the authors address some of my concerns.

Main comments/questions:
- In the end, it seems like solving the problem of the shift in the mean is the main thing that makes SELU, sSELU and lSELU work on MobileNets. Is there a problem of gradient explosion in SELU apart from growing means that sSELU or lSELU solves, and if so, what is an experiment that shows this? Without this, it is not clear to me whether the slightly better results of sSELU and lSELU is simply a result of sub-optimal hyperparameter tuning of the baselines (see comment below).
- The BN+ReLU baseline run on MobileNet seems a bit suboptimal to me. The original MobileNet paper reports 71.7% accuracy, and other work reports even higher numbers, e.g., 72% in https://arxiv.org/pdf/1710.05941.pdf. This could be a result of the learning rate not being tuned for any of the activation functions. It is not obvious to me that the optimal learning rate should be the same across all activation functions, so ideally optimal performance should be shown over a learning rate sweep for each method.
- The authors mention that neural networks with larger fan-in are more likely to lose the self-normalization effect on the mean based on the observation that if the mean is less than 1/N, then multiplication with the weight decreases the mean of the pre-activations. I do not however follow this argument (and don't think it is true), since the weights are typically initialized from N(0, 1/N), and therefore the probability that the mean is less than 1/N does not change with larger N?
- How sensitive is performance to epsilon? The fact that epsilon needs to be tuned seems to be a major disadvantage of the proposed activation functions.
- It would be interesting to see the following ablations as well for the MobileNet problem:
    - SELU + Weight Centralization
    - dSELU + Mixup
    - dSELU + Weight Centralization

Other minor comments:
- It would have been good to report the values of the parameters alpha, beta and lambda of sSELU and lSELU for at least one problem (such as the MobileNet experiment), and compare them with SELU.
- Would be good to add SELU in figures 1b, 1d and 3.
- The authors mention in definition 2 that the self-normalization property is stronger when phi(q) is closer to 1/q. Can the authors expand on why this is? It is not obvious to me from the definition.

Additional comments on the writing:
- It is not clear to me how the new definition is "easier to use both analytically and numerically".
- Explanation of related prior work should be improved, as it is not quite clear right now. In particular, the explanation of the prior work in mean field theory, as well as the explanation of "gradient norm equality".
- The first paragraph on the second page needs to be checked for a number of typos (Forbenius -> Frobenius, etc) and grammatical errors.
- Typo at the end of page 2: boarder -> border
- Equation 13 and 15 need to be made clearer by adding "if q_l > 1", etc.

----------

Update after rebuttal: I thank the authors for the detailed response. Some of my comments have been addressed. However, some of my major concerns remain such as the insufficiency of the hyperparameter tuning for the baselines to do a fair comparison. I am keeping my score.

---

> ### Author Response · Authors · 2020-11-24
> **Response to Reviewer 1 (Part 1)**
>
> Thanks for your constructive feedback. We have carefully revised our paper and enriched the experiments as much as possible.
>
> **(1) In the end, it seems like solving the problem of the shift in the mean is the main thing that makes SELU, sSELU, and lSELU work on MobileNets. Is there a problem of gradient explosion in SELU apart from growing means that sSELU or lSELU solves, and if so, what is an experiment that shows this? Without this, it is not clear to me whether the slightly better results of sSELU and lSELU is simply a result of sub-optimal hyperparameter tuning of the baselines (see comment below).**
>
> **Response**: Our experiments are organized in two parts. On one hand, the Section 6.2 of the revised paper focuses on the gradient problem with experiments on a 56-layer CNN. On the other hand, as the MobileNet V1 only has 16 layers, the gradient explosion of second moment is less severe. And the growing mean becomes the major obstacle that stops us from using these activation functions on ImageNet. As a result, the experiments on MobileNet V1 - ImageNet is majorly used to demonstrate the effectiveness of the techniques we proposed in the Section 5: Large-Scale Self-Normalizing Neural Network of the revised paper, in which we theoretically show why these techniques are effective.
>
> **(2) The BN+ReLU baseline run on MobileNet seems a bit suboptimal to me. The original MobileNet paper reports 71.7% accuracy, and other work reports even higher numbers, e.g., 72% in** [**https://arxiv.org/pdf/1710.05941.pdf**](https://arxiv.org/pdf/1710.05941.pdf)**. This could be a result of the learning rate not being tuned for any of the activation functions. It is not obvious to me that the optimal learning rate should be the same across all activation functions, so ideally optimal performance should be shown over a learning rate sweep for each method.**
>
> **Response**: It is true that our BN + ReLU baseline is little bit lower than the result in the original paper. However, this is majorly due to two reasons.
>
> First, the original paper doesn't clarify how many epochs are used as well as the detailed learning rate decay strategy. However, the script at https://github.com/tensorflow/models/tree/master/research/slim/nets/mobilenet  shows that training MobileNet v2 on a single GPU takes 5.5M steps under batch size 96, which is roughly 400 epochs. And it is somehow reasonable to infer than MobileNet v1 takes the same strategy. However, 400 epochs is enough for methods with different convergence rate to converge to the similar accuracy. Therefore, we follow the training scheme used in [a] which only takes 90 epochs, which allows us to somehow reflect the convergence rate of different methods.
>
> Second, the 72% accuracy in [https://arxiv.org/pdf/1710.05941.pdf](https://arxiv.org/pdf/1710.05941.pdf) is based on RMSProp, which adjusts the update of the parameters by dividing the moving average of the square root of the Mean Square of the gradient. This has similar effect with the self-normalization respective to the gradient norm equality. Therefore, to better illustrate the stronger self-normalization property provided by lSELU and sSELU, we use the simple SGD with momentum as the optimizer.
>
> For learning rate sweeping, one of the advantage brought by batch normalization is that it allows the use of arbitrary learning rate. Therefore, as the activation functions with the self-normalization property is a proxy of BN, instead of sweeping the learning rate for better performance, we use an arbitrary learning rate 0.02 for fair comparison.
>
> **(3) The authors mention that neural networks with larger fan-in are more likely to lose the self-normalization effect on the mean based on the observation that if the mean is less than 1/N, then multiplication with the weight decreases the mean of the pre-activations. I do not however follow this argument (and don't think it is true), since the weights are typically initialized from N(0, 1/N), and therefore the probability that the mean is less than 1/N does not change with larger N?**
>
> **Response**: It is true that the probability that the mean is less than $1/N_{l-1}$ decreases when $N_{l-1}$ get larger. However, in the Appendix A.5 of the revised paper, we empirically show that it decreases under a lower speed compared with the increase of $N_{l-1}$, and networks with larger fan-in tend to have larger $\mu N_{l-1}$, which implies weaker self-normalization property.

---

> > ### Author Response · Authors · 2020-11-24
> > **Response to Reviewer 1 (Part 2)**
> >
> > **(4) How sensitive is performance to epsilon? The fact that epsilon needs to be tuned seems to be a major disadvantage of the proposed activation functions.**
> >
> > **Response**: In the revised paper, instead of just reporting the best performance, we report the result under all the $\epsilon$ in Table 1 of Section 6.2 in the revised paper. Unlike [a] that illustrate their result on a 32-layer network, we choose 56 layers to better demonstrate the strength of self-normalization property. However, with more layers, the trade-off between self-normalization property and speed of gradient explosion gets too complex to be simply captured with $\epsilon \approx 1/L$. Actually, as shown in Table 1, even with dSELU in [a], the best accuracy is achieved with $\epsilon > 1/L$.
> >
> > Although tuning the $\epsilon$ may be less convenient, Table 1 shows that arbitrarily taking $\epsilon \approx 1/L$ still achieves consistent performance improvement over SELU [b]. Besides, under the same $\epsilon$, our sSELU and lSELU achieve approximately the same or even better performance than dSELU [a], especially for sSELU on CIFAR-10 and CIFAR-100, lSELU on Tiny ImageNet.
> >
> > Last but not least, we add some additional discussion on choosing $\epsilon$ in the last paragraph of Section 4.
> >
> > **It would be interesting to see the following ablations as well for the MobileNet problem:**
> >
> > * **SELU + Weight Centralization**
> >
> > * **dSELU + Mixup**
> > * **dSELU + Weight Centralization**
> >
> > **Response**: We have enriched the experiments on MobileNet in the revised paper including these three experiments. Besides, a new configuration, Centralized Activation function, is proposed in Section 5 and evaluated (marked with central in Table 2). This new configuration achieves higher accuracy (e.g. 71.95% for sSELU). The theoretical part for this new configuration can be found at the end of Section 5 of the revised paper. Notably, the centralized activation function is not applicable in SELU and dSELU, as they don't have enough parameters.
> >
> > **(5) It would have been good to report the values of the parameters alpha, beta and lambda of sSELU and lSELU for at least one problem (such as the MobileNet experiment), and compare them with SELU.**
> >
> > **Response**: In the revised paper, we summarize the values of parameter $\alpha$, $\lambda$, $\beta$, and the corresponding $\gamma_{q=1}$ in Table 5, Appendix D. The $\gamma_{q=1}$ is a value that characterizes the strength of self-normalization property (Proposition 3.1). Notably, smaller $\gamma_{q=1}$ indicates stronger self-normalization property.
> >
> > **(6) Would be good to add SELU in figures 1b, 1d and 3.**
> >
> > **Response**: Figure 1 is ploted under $\epsilon=0.03$.  Whereas SELU doesn't has such option, therefore it is not included in the figure. Beside, the major purpose of Figure 1 is to illustrate that sSELU and lSELU can be configured to get closer to $1/q$. For Figure $3$, as SELU is a special case of dSELU under $\epsilon\approx 0.07$, the curve of dSELU under $\epsilon=0.07$ can be viewed as the curve of SELU.
> >
> > **(7) The authors mention in definition 2 that the self-normalization property is stronger when phi(q) is closer to 1/q. Can the authors expand on why this is? It is not obvious to me from the definition.**
> >
> > **Response**: We formulize the strength of self-normalization property in Proposition 3.1 of the revised paper. Let $q$ be the second moment of the input pre-activation of the activation function. This time we view $\phi(q)$ as an interpolation between $1$ and $1/q$ under a factor $\gamma_q$. For any $q$, $\gamma_q$ goes to $0$ when $\phi(q)$ is closer to $1/q$.  In Appendix A.2, we theoretically show that $\gamma_q$ determines the speed that the statistics in both forward and backward passes converging toward their fixed points. Moreover, we show that when $q\approx 1$, $\gamma_q$ can be approximated with $\frac{d\phi(q)}{dq}|_{q=1}+1$, which is the object function to minimize when solving the $\lambda$, $\alpha$, and $\beta$ of lSELU and sSELU. (Eq. 11 of the revised paper).
> >
> > **(8) It is not clear to me how the new definition is "easier to use both analytically and numerically".**
> >
> > **Response**: The argument "easier to use both analytically and numerically" is respective to the original Definition 1 proposed by [b]. While the original definition simply describes the desired behavior of a self-normalizing neural network, it is hard to be used to exam whether individual activation function has such property, not to mention comparing the strength of self-normalization property between two activation functions.
> >
> > Oppositely, our Definition 2 defines the Self-normalization Property with Eq. 6 and Eq. 7 in the revised paper, these two equations can be easily examined both analytically and numerically.

---

> > > ### Author Response · Authors · 2020-11-24
> > > **Response to Reviewer 1 (Part 3)**
> > >
> > > **(9) Explanation of related prior work should be improved, as it is not quite clear right now. In particular, the explanation of the prior work in mean field theory, as well as the explanation of "gradient norm equality".**
> > >
> > > **Response**: In the revised paper, we formulize the forward and backward propagtion as Proposition A.1 and Proposition A.2 in the Appendix. More detailed derivations are provided along with the refered definitions and theorems. The "gradient norm equality" means that the Frobenius norm of the gradient is more or less equal in different layers, so that the information flow in backward pass can be well preserved. This is clarified at the end of Sec. 2 of the revised paper.
> > >
> > > **(10) The first paragraph on the second page needs to be checked for a number of typos (Forbenius -> Frobenius, etc) and grammatical errors; Typo at the end of page 2: boarder -> border; Equation 13 and 15 need to be made clearer by adding "if q_l > 1", etc.**
> > >
> > > **Response**: We have fixed the typos in the revised paper. Thank you for your kind reminder.
> > >
> > > #### References
> > >
> > > * [a] Z. Chen, L. Deng, B. Wang, G. Li and Y. Xie, "A Comprehensive and Modularized Statistical Framework for Gradient Norm Equality in Deep Neural Networks," in IEEE Transactions on Pattern Analysis and Machine Intelligence, doi: 10.1109/TPAMI.2020.3010201.
> > > * [b] Klambauer, G., Unterthiner, T., Mayr, A., & Hochreiter, S. (2017). Self-normalizing neural networks. In *Advances in neural information processing systems* (pp. 971-980).

---

### Official Review · AnonReviewer4 · 2020-10-29
**There seem to be multiple gaps in the theory and the modifications do not mitigate gradient explosion**

**Rating:** 4
**Confidence:** 5

**Review:**

## Summary

The paper proposes two modifications to SELU activation function to improve it with regards to preserving forward-backward signal propagation in neural networks. The work builds on top of the mean-field theory literature and provides a modified self-normalization property (additional constraints compared to SELU). Further, it discusses some heuristics (mixup, weight centralization) to improve performance in practice.

## Strengths

1. The problem is interesting and important to improve trainability in neural networks. It is shown recently that SELU suffers from gradient explosion and this work attempts to circumvent it by redefining the self-normalization condition. This condition leads to one more scalar parameter in SELU function and discusses a method to obtain this scalar.

2. Experiments are conducted on a 56-layer CNN for cifar and tinyimagenet datasets and modified mobilenetv1 for imagenet. These proposed modifications to SELU seem to result in improved accuracy even though the improvement is minimal on imagenet.

3. Overall the paper is clearly written.

## Weaknesses

The main weaknesses of the manuscript in my opinion are as follows:

1. There seem to be multiple gaps in the theory and the modifications do not mitigate gradient explosion:
	- The theoretical analysis lacks rigor and the main purpose of mitigating the gradient explosion issue of SELU is not sufficiently addressed. In fact, if I understand correctly, even with the redefined self-normalization property gradient vanishing/explosion can occur. Note, according to Eq. 15, when $q_l <1$, the numerator can grow arbitrarily. Please clarify.
	- Related to the above point, the 3rd condition in Eq. 12 and the subsequent argument of gradient norm converges to the fixed point is unsubstantiated. To my understanding, Eq. 15 only gives a lower/upper bound depending on the value of $q$ and it does not convey anything regarding convergence to such a fixed point. If it does a rigorous proof would be required.
	- The assumption in Eq. 4 is unjustified. In Fig. 3 there is a plot provided but it has several underlying assumptions (eg, small $\epsilon$ and $E[x^2]$ etc.) and it is not clear what functions would satisfy this assumption. I recommend the authors to look at Gaussian-Poincare inequality for connecting a function and its derivative under the Gaussian distribution. In fact, GP inequality is recently used to define a class of activation functions to solve gradient vanishing/explosion in neural networks [a].
	- The condition on $\lambda$ (Eq. 19) is derived without any theoretical basis and it seems like it is found empirically. This also weakens the theoretical contribution of the paper.

2. Contradicting arguments compared to dynamical isometry and mean-field theory literature:
	- In dynamical isometry and mean-field theory literature it is shown that dynamical isometry is sufficient to ensure stable gradients and for RELU the scalar is computed to be $\sqrt{2}$. Refer [b]. To this end, it is confusing to me when it is mentioned that this condition "will lose self-normalization" in the second last para of page 4. Please precisely define what is meant by self-normalization (including its purpose) in this paper and also clarify this confusion.
	- In the literature it is known that when the width is large, the preactivations will be close to a 0 mean Gaussian distribution due to the central limit theorem (Assumption 2 in this paper). However, in 3rd last para of page 7, it is mentioned that "networks with large fan-in are more likely to lose the self-normalization effect". This seems to be contradicting as well.

3. Discussion about the mean of activations:
	- The discussion about the mean of activations exploding in Sec. 6 seems irrelevant to me. In the theoretical derivation of this paper, I could not find any discussion on the mean of activations affecting any of the desired quantities such as $q$ or $\phi(q)$. Please improve the clarity on how this is connected to the theory of this paper.

4. No experiments on gradient vanishing or exploding behaviour:
	- As far as I understood, the main motivation is to fix the gradient exploding issue of SELU. However, there is no experiment showing this effect and this seems to reinforce my concern that the proposed modification does not solve the gradient vanishing/exploding issue (also mentioned briefly in the para before conclusion). This questions the significance of the contributions.
	- I understand that generalization error is the quantity that we mostly care about. However, self-normalization or the work in mean-field theory literature concerns on trainability. Therefore, I believe, it is important to show the trainability behaviour and the propagation of gradients. Note, the theory does not convey anything about the generalization error directly but the signal propagation in the forward and backward directions.

In retrospect, I just want to say that in the current form theory and heuristics are mixed together making it difficult to see where the benefit is coming from and I think if theory is tightened this could be a good paper.

## Minor Comments

1. It seems the modifications could be done to other activation functions as well such as tanh etc. Please consider.
2. Please explain what is the meaning of $\epsilon$ in Eq. 10. This is not clear from this paper.
3. First para in Sec. 7.1 : "considerably higher"

## References

- [a] Lu, Y., Gould, S. and Ajanthan, T., 2020. Bidirectional Self-Normalizing Neural Networks. arXiv.
- [b] Pennington, J., Schoenholz, S. and Ganguli, S., 2017. Resurrecting the sigmoid in deep learning through dynamical isometry: theory and practice. NeurIPS.

---

> ### Author Response · Authors · 2020-11-24
> **Response to Reviewer 4 (Part I)**
>
> We sincerely appreciate your valuable comments. We have carefully revised the paper and the responses are listed as follows.
>
> **(1) The theoretical analysis lacks rigor and the main purpose of mitigating the gradient explosion issue of SELU is not sufficiently addressed. In fact, if I understand correctly, even with the redefined self-normalization property gradient vanishing /explosion can occur. Note, according to Eq. 15, when $q_l<1$,  the numerator can grow arbitrarily. Please clarify**
>
> **Response**: The Frobenius norm of gradient in the original Eq. 15 converges toward the fixed point if you view it in the reversed perspective, i.e. the gradient propagates from the input layer to the output layer. Then, Eq. 15 shows that $E[||\frac{\partial \mathcal{L}}{\partial x_l}||_2^2]$ is closer to $q_0E[||\frac{\partial \mathcal{L}}{\partial h_0}||_2^2]$ compared with $E[||\frac{\partial \mathcal{L}}{\partial h_l}||_2^2]$, where $x_l=f(h_l)$.
>
> Our analysis of the backward pass is based on the conclusions in [a], in which the evolution of Frobenius norm of the backward gradient in layer $l$ can be described by multiplying with a scalar $tr(J_l^TJ_l)$, where $tr$ is the normalized trace and $J_l$ is the Jacobian matrix of the layer $l$. This allows us to view the back propagation in the reversed perspective as we only care about the Frobenius norm.
>
> Viewing the back propagation in the reversed perspective greatly simplifies the analysis. On one hand, the $tr(J_l^TJ_l)$ of activation function is the function of the second moment of the input pre-activation, which is only determined by its preceding layers. On the other hand, under the reversed perspective of backward pass, the Frobenius norm of layer $l$ is also determined only by the preceding layers. As a result, we only have to study the preceding layers of layer $l$.
>
> **(2) Related to the above point, the 3rd condition in Eq. 12 and the subsequent argument of gradient norm converges to the fixed point is unsubstantiated. To my understanding, Eq. 15 only gives a lower/upper bound depending on the value of $q$ and it does not convey anything regarding convergence to such a fixed point. If it does a rigorous proof would be required.**
>
> **Response**: Thank your for your insightful comment. Following your suggestion, we formulize the strength of self-normalization property in Proposition 3.1 of the revised paper. Let $q$ be the second moment of the input pre-activation of the activation function. This time we view $\phi(q)$ as an interpolation between $1$ and $1/q$ under a factor $\gamma_q$. For any $q$, $\gamma_q$ goes to $0$ when $\phi(q)$ is closer to $1/q$.  In Appendix A.2, we theoretically show that $\gamma_q$ determines the speed that the statistics in both forward and backward passes converging toward their fixed points. Moreover, we show that when $q\approx 1$, $\gamma_q$ can be approximated with $\frac{d\phi(q)}{dq}|_{q=1}+1$, which is the object function to minimize when solving the $\lambda$, $\alpha$, and $\beta$ of lSELU and sSELU. (Eq. 11 of the revised paper).
>
> **(3) The assumption in Eq. 4 is unjustified. In Fig. 3 there is a plot provided but it has several underlying assumptions (eg, small $\epsilon$ and $E[x^2]$ etc.) and it is not clear what functions would satisfy this assumption. I recommend the authors to look at Gaussian-Poincare inequality for connecting a function and its derivative under the Gaussian distribution. In fact, GP inequality is recently used to define a class of activation functions to solve gradient vanishing/explosion in neural networks.**
>
> **Response**: Thank you for your kind recommendation. However, in the revised paper, we find it is better to remove the original assumption in Eq. 4 and replace it with $(1+\delta_q) E[f^2(x)] = E[(df(x)/dx)^2]E[x^2]$, where $x\sim N(0, q)$ (Eq. 2 of the revised paper). The $\delta_q$ is a function of $q$. In Proposition 3.1, we show that $1 + \delta_q$ determines the speed of gradient explosion in the backward pass. In Fig. 2, we further show the relationship between $(1+\delta_q)$ and $q$, $\epsilon$.
>
> Considering the nonzero $\delta_q$ allows us to provide more insight on the selection of $\epsilon$ (Last paragraph of Section 4) as well as additional theoretical insight behind the Mixup data Augmentation. (Section 5). Last but not least, this result accords with the experiment in Section 6.1 of the revised paper, in which we plot the distribution of the second-moment of forward pre-activation and Frobenius norm of backward gradient under different activation functions.

---

> > ### Author Response · Authors · 2020-11-24
> > **Response to Reviewer 4 (Part 2)**
> >
> > **(4) The condition on $\lambda$ (Eq. 19) is derived without any theoretical basis and it seems like it is found empirically. This also weakens the theoretical contribution of the paper.**
> >
> > **Response**: This constraint is first proposed in the original SELU paper [b] as follows: "a slope larger than one to increase the variance if it is too small in the lower layer."
> >
> > In the revised paper, besides refering to this argument, we provide two additional theoretical insights. First, we show that having $\lambda\approx 1$ helps maintain the mean of the output pre-activations around 0, which is helpful under large learning rate and fan-in. Second, we show that larger $\lambda$ reduces $\delta_q$, thus it slows down the gradient explosion in the backward pass. These two insights together justify the constraint $\lambda \ge 1$. The detailed discussion can be found in Appendix A.4 of the revised paper.
> >
> > **(5) In dynamical isometry and mean-field theory literature it is shown that dynamical isometry is sufficient to ensure stable gradients and for RELU the scalar is computed to be $\sqrt 2$. To this end, it is confusing to me when it is mentioned that this condition "will lose self-normalization" in the second last para of page 4. Please precisely define what is meant by self-normalization (including its purpose) in this paper and also clarify this confusion.**
> >
> > **Response**: The self-normalization property in our paper means that the deviation of the statistics in both forward and backward passes can be gradually fixed during propagation.
> >
> > Let's consider training a deep neural network under relatively large learning rate. During training, the nice properties like entires in the weight matrix following $N(0, 1/fan\_in)$ are not likely to hold, thus it is very likely that the statistics of forward and backward signals deviate from the fixed point in some layers. RELU with scalar $\sqrt 2$ doesn't respond to such deviation, the best it can do is keeping the input and output to have the same second moment. On the other hand, activation functions like SELU and our sSELU, lSELU are sensitive to the second moment of the input pre-activations. Once the statistics deviate from the fixed point in one layer, all the succedding layers will work together to push the statistics toward the fixed point until the deviation is fixed. Therefore, the latter one is more robust to larger learning rate, network depth, and even some arbitrary assumptions. We have clarified it in the Introduction section of the revised paper.
> >
> > **(6) In the literature it is known that when the width is large, the preactivations will be close to a 0 mean Gaussian distribution due to the central limit theorem (Assumption 2 in this paper). However, in 3rd last para of page 7, it is mentioned that "networks with large fan-in are more likely to lose the self-normalization effect". This seems to be contradicting as well.**
> >
> > **Response**: It is true that under large width, the output pre-activation converges to Gaussian distribution under central limit theorem. However, the "0 mean" only holds if either the input activation or the entries in the weight have 0 mean, which is not necessarily held during training.
> >
> > Our argument in the original 3rd last para of page 7 tries to capture the situation in which neither these two conditions hold. In the revised paper, we formulize it as Proposition 3.2. Besides, in the Appendix A.5 of the revised paper, we empirically show that networks with larger fan-in tend to have larger $\mu N_{l-1}$, which implies weaker self-normalization property.
> >
> > **(7) The discussion about the mean of activations exploding in Sec. 6 seems irrelevant to me. In the theoretical derivation of this paper, I could not find any discussion on the mean of activations affecting any of the desired quantities such as $q$ or $\phi(q)$. Please improve the clarity on how this is connected to the theory of this paper.**
> >
> > **Response**: the self-normalization property is derived based on the Assumption 1, which indicates that the input of the activation function has 0-mean. However, under large fan-in, the assumption on 0-mean input is likely to be violated. We have clarified this point in the end of the second paragraph of Section 5.

---

> > > ### Author Response · Authors · 2020-11-24
> > > **Response to Reviewer 4 (Part 3)**
> > >
> > > **(8) As far as I understood, the main motivation is to fix the gradient exploding issue of SELU. However, there is no experiment showing this effect and this seems to reinforce my concern that the proposed modification does not solve the gradient vanishing/exploding issue (also mentioned briefly in the para before conclusion). This questions the significance of the contributions.**
> > >
> > > **Response**: In the Section 6.1 of the revised paper, we plot the distribution of the second-moment of forward pre-activation and Frobenius norm of backward gradient on the weights in 10 training epochs under relatively large learning rate. The result shows that our sSELU and lSELU has flatter and more concentrated distribution in both forward and backward passes compared with SELU and dSELU, which demonstrates that our sSELU and lSELU have stronger self-normalization property.
> > >
> > > **(9) I understand that generalization error is the quantity that we mostly care about. However, self-normalization or the work in mean-field theory literature concerns on trainability. Therefore, I believe, it is important to show the trainability behaviour and the propagation of gradients. Note, the theory does not convey anything about the generalization error directly but the signal propagation in the forward and backward directions.**
> > >
> > > **Response**: While the propagation of both forward activations and backward gradient are shown in Fig. 4 of Section 6.1 of the revised paper, the results in Section 6.3 also show that all the techniques proposed in Section 5 solve the gradient explosion problem on large SNNs like MobileNet V1. For example, adding Mixup Data Augmentation makes SELU trainable. This is theoretically explained in Section 5 Mixup Data Augmentation.
> > >
> > > Besides, we add additional discussion on the regularization of the training process in Section 5 of the revised paper. For example, using Mixup Data Augmentation provides additional regularization power.
> > >
> > > **(10) It seems the modifications could be done to other activation functions as well such as tanh etc. Please consider.**
> > >
> > > **Response**: Thanks for your suggestion. In our paper, we only use lSELU and sSELU as two demos. In future work, the AutoML techniques can be used, and our theorems can be used to quickly exam the candidate activation functions without actually running the experiments.
> > >
> > > However, one benefit brought by the ELU-shape activation functions is that it has linear positive part, which reduces the $\delta_q$ and has low gradient explosion rate (Eq. 39 of the revised paper). On the other hand, has mentioned in [a], tanh may lead to large $1+\delta_q$. Therefore, it may be less favorable.
> > >
> > > **(11) Please explain what is the meaning of ϵ in Eq. 10. This is not clear from this paper.**
> > >
> > > **Response**: The hyper-parameter $\epsilon$ is first introduced in [a], which is used to control the trade-off between the strength of self-normalization property and the speed of gradient explosion. Actually, the original SELU [b] has $\epsilon\approx 0.0716$ even though it is not explicitly determined. In this paper, we use the same $\epsilon$ as [a]. The related discussion can be found under Eq. 5 of the revised paper.
> > >
> > > **(12) First para in Sec. 7.1 : "considerably higher"**
> > >
> > > **Response**: We have fixed it in the revised paper. Thank you for your kind reminder.
> > >
> > > #### Reference
> > >
> > > * [a] Z. Chen, L. Deng, B. Wang, G. Li and Y. Xie, "A Comprehensive and Modularized Statistical Framework for Gradient Norm Equality in Deep Neural Networks," in IEEE Transactions on Pattern Analysis and Machine Intelligence, doi: 10.1109/TPAMI.2020.3010201.
> > > * [b] Klambauer, G., Unterthiner, T., Mayr, A., & Hochreiter, S. (2017). Self-normalizing neural networks. In *Advances in neural information processing systems* (pp. 971-980).

---

### Decision · Program_Chairs · 2021-01-07
**Final Decision**

**Decision:**

Reject

**Comment:**

This paper proposed two variants of the SELU activation function, termed the leaky SELU (lSELU) and scaled SELU (sSELU), respectively, in order to yield a stronger self-normalization property. The review process and the discussion find the following issues:

- The hyperparameter tuning for the baselines is insufficient for the baselines so that the comparison may be unfair.
- The experiment results (Table 2) do not show superiority of the proposed activation functions. In addition, the results appear to be unrelated to each other. (see Reviewer 3's detailed update)
- Reviewer 2 pointed out that the architecture that the authors used was far from the SOTA. I read the authors' response. This paper may benefit from adding some even naive workaround and making fair comparisons under the SOTA architecture.

I do not think (6) is a good way to present this equation. The authors may want to perform change of variable and replace $\sqrt{q}z$ by $z$ in the integral and add this form to the right-hand side of (6). In addition, $\epsilon$ appears in Definition 2. However, when the authors mention the self-normalization property in Definition 2, they omit $\epsilon$. It might be better to call it $\epsilon$-self-normalization property to stress that this definition depends on $\epsilon$.

Other minor issues:
- The line right below (15) on page 11, the authors did not need to capitalize "orthogonal".
- In eq (16), $W_l^{,T}$, the comma is unnecessary.